# Comparison performance of the Bayesian Approach with the Weibull and Birnbaum-Saunders distributions in imputation of time-to-event censors

**Parviz Shahmirzalou[1], Aliakbar Rasekhi[1]\*, Majid Jafari Khaledi[2], Maryam Khayamzadeh[3]\***

**1** Department of Biostatistics, Faculty of Medical Sciences, Tarbiat Modares University, Tehran, Iran, **2** Department of Statistics, Tarbiat Modares University, Tehran, Iran, **3** Cancer Research Center, Shahid Beheshti University of Medical Sciences, Tehran, Iran

\* rasekhi@modares.ac.ir (AR); khayamzadeh@yahoo.com (MK)

**Data Availability Statement:** The code and Breast cancer data have been uploaded to GitHub Link for

## Abstract

Almost all survival data is censored, and censor imputation is necessary. This study aimed to investigate the performance of the Bayesian Approach (BA) in the imputation of censored records in simulated and Breast Cancer (BC) data. Due to the difference in the distribution of time to event in survival analysis, two well-known the Weibull and Birnbaum-Saunders (BS) distributions have been used to test the performance of the BA. For each of the censored, 10,000 times were simulated using the BA in R and BUGS software, and their median or mean was imputed instead of each censor. The eligibility of both imputation methods was investigated using different curves, different censoring percentages, and sample sizes, as well as the Deviance Information Criteria (DIC), Effective Sample Size, and the Geweke diagnostic in simulated and especially real BC data. The BC data, which contains 220 patients who were identified and followed up between 2015 and 2023, was made accessible on February 1, 2023. The Kaplan-Meier, the BA, and other survival curves were drawn for the observed times. Findings indicated that the performance of the BA under the Weibull and BS distributions in simulated data is similar. The DIC index in the BC data under the BS distribution (1510) is less than the Weibull distribution (1698). Therefore, the BS distribution is preferred over the Weibull for imputation of censoring times in real BC data.

## 1- Introduction

There are several uses for survival statistics across many disciplines, particularly in the field of medicine. The analysis of survival data is crucial, especially when it pertains to human life. This study is different from other statistical methods because it looked at how long a patient lived and how they ended up. Although the definition of the term "survival" varies depending on the context in which it is used, in the field of medical sciences, the term "survival" refers to period of time before an event (such as death, a recurrence, a relapse, a metastasis, etc.) occurs.

R codes: https://github.com/pshkhoei/Thesis-Paper-1.

**Funding:** The author(s) received no specific funding for this work.

**Competing interests:** The authors have declared that no competing interests exist.

This period of time (survival time) can take the form of hours, days, weeks, etc. The aforementioned traits have motivated numerous researchers to study survival [1]. In a survival study, patients are monitored for a certain period of time; some of them experience the desired event while the study is ongoing, while others do not experience the target event at all. These patients are known as censored, and it is unknown how long they will live after being censored. People are censored for a variety of reasons, including ending the research, leaving it, experiencing the event inside a particular time window, and a few more. As a result, censoring is divided into various categories, including left, right, and interval, which complicates survival analyses [2]. It is improper to remove the censored records. When describing the data and making predictions based on the model, it is required to impute the missing data. Some statistical techniques require imputation. Missing data imputation is done to maximize the use of available information. So far, different methods have been proposed for imputation (average, median, single, and multiple imputation), each of which has its strengths and weaknesses [3–6]. However, the Bayesian approach (BA) has some advantages for imputing incomplete datasets, including high efficiency in low sample sizes, the ability to handle heavy censoring, and the ability to assign prior distributions to parameters [7, 8]. Consequently, a complex approach to the imputation of those missing values is developed to accurately accommodate incomplete data. To impute missing data with plausible values, numerous machine learning algorithms have been proposed [9, 10].

In Part 2, we will discuss the several imputation techniques that have been proposed so far. Survival analysis frequently uses the Weibull distribution [1, 2], and the Birnbaum-Saunders (BS) distribution is viewed as the Weibull distribution's main competitor [11]. In Section 3, the Weibull and BS distributions are introduced. It is possible to use frequentist or BA for imputation, and in Section 4 the parametric BA is introduced for the Weibull and BS distributions. This section also discusses the mean and median of the distributions used in this approach, as well as how they are applied to the distributions in question. The steps that must be taken to perform the BA based on the Weibull or BS distributions, simulation of the data, and finally imputation of censors is discussed in Section 5, and are drawn in Sections 5–2 and 5–3. Section 5.4 applies the procedures outlined in Section 5 to data on Breast Cancer (BC) and presents the results, keeping in mind that the effectiveness of any novel method should be evaluated using real data. We'll discuss and come to a conclusion in Sections 6 and 7, respectively. The primary goal of this study is to evaluate the effectiveness of the BA for imputed censored times using two applicable distributions.

## 2- Literature review on imputation

As technology advances, the number of records grows every day. To analyze data, it is important to use the appropriate techniques to fill in missing or incomplete data. So far, many different studies have been done on how to fill in the missing information. Tanner and Wong presented a method for estimating the nonparametric hazard function in censored and grouped survival data. The failure times were grouped and allocated a particular distribution for this purpose. The imputation distribution is used to compute the failure times in each iteration of the procedure. Cross-validation is used in non-parametric likelihood to estimate the smoothed size of the parameters. After obtaining an estimate of the hazard function and estimating the variance of the estimate by repeating the imputation procedure [12].

Some researchers have employed multiple imputations. Multiple imputation is a statistical technique for handling missing data that involves numerous calculations. Rubin was the first person to come up with the idea of multiple imputation in 1987. To remove any doubt about what should be imputed, this approach imputes each missing data point many times. Multiple

imputation, according to Rubin, offers two key benefits: first, it makes possible the use of analytical techniques that call for complete data, and second, it allows for the application of knowledge in the presence of incomplete data [13]. Satten et al. assessed censored times in a study by assigning distributions to times and estimating distribution parameters using Proportional Hazard (PH) regression [14]. Schaubel and Cai explored the imputation of lost times in survival studies of the kind of recurring events that are highly common in clinical and epidemiological studies. Because both the survival duration and the type of event are absent in these studies, imputation presents a unique set of challenges. Two missing imputation techniques are suggested in this study. To employ these techniques, random censoring is required. The authors of this research have indicated that the proposed procedures are valid [15]. The multiple imputation necessary presumptions have been established by Newgard et al. They have mentioned random or completely random censorship, and the randomness of censorship has been confirmed by a large number of studies. It is also said that censoring must be done at random for imputation methods based on likelihood and the BA. Although it was stated in the research that multiple imputation was robust to this assumption, some studies have specified having a multivariate normal distribution as one of the assumptions of multiple imputation [16].

A novel approach to Bayesian network-based missing data imputation has been presented by Niloofar and Ganjali [17]. In this study, imputation is done with information about the Markov blanket and the parents of each node in the Bayesian network [17]. In a paper, Pratama et al. [18] explored many approaches to handling missing data, particularly time series data. In this study, it was argued that the normal ways of filling in missing data, like imputation with mean, median, and case deletion, are not very reliable. Instead, it was suggested that multiple imputation methods be used. In this way, the authors talked about the genetic algorithm method, the fuzzy method, the maximum probability method, and so on. The research's authors suggested that the imputation approach be chosen depending on the type of investigation [18]. For bounded variables, multiple imputation has been studied by Geraci and McLain. Based on this study, most imputation methods are used for infinite continuous variables and with corrections for limited variables, which could lead to inappropriate results. For variables with a bounded distribution, a multiple imputation method based on quartiles is suggested as a result. The authors of this study believe that the advantages of the suggested strategy include low bias and little variance between imputations [4].

The Cox model is based on the assumption of proportional hazards. The validity of the proportional hazard assumption in the analysis of ordinal data has been surveyed. The method that is being described for missing and repeated data is based on multiple imputations. The validity of the proportional hazard assumption was checked for different amounts of censoring and by taking into account the first-type error and the power of the test [19].

Erler has suggested methods for coping with missing data in both simple and complicated circumstances in an article. According to one of them, 3 to 5 repeats for each missing data point are required for multiple imputations to be reliable. It was additionally noted that the use of covariates in predicting the amount of missing data is advised to improve the imputation's accuracy. The number of covariates can range from 20 to 30 variables, based on the size of the study sample and the hardware facilities [5]. The imputation of missing data in a cross-over experiment with a time-to-event outcome was investigated. Because of the unique design of the cross-over study, the sample size has been drastically reduced due to missing data. In this study, covariates are used to fit a model based on the median residual lifetime. In a two-stage crossover trial, the lost times from the first period are imputed first, followed by the lost times from the second phase [5]. Turkson et al. have conducted a systematic review of various methods of missing data imputation. According to this study, there are new imputation methods such as censored network estimation, imputation based on conditional average,

imputation based on reverse weighting, and a few others, in addition to the traditional imputation methods like complete data analysis, imputation approach, data dichotomization, and likelihood-based approach. The authors have concluded that the correct method of imputation of missing values should be chosen after carefully examining the type of missing value and the type of data structure [20].

## 3- The Weibull and Birnbaum-Saunders distributions

### 3-1- The Weibull distribution

Due to its tractability, high degree of flexibility, and availability in most statistical tools, the Weibull distribution is frequently employed in the analysis of survival data. Following are the functions of the Weibull distribution's density (f), survival (S), and hazard (h) for times $t_i$ with the shape ($\alpha$) parameter and scale ($\beta$) parameter [21].

$$t_i \sim Weibull(\alpha, \beta) \tag{1}$$

$$f(t|\alpha, \beta) = \alpha\beta t^{\alpha-1}e^{-\beta t^\alpha}$$

$$h(t|\alpha, \beta) = \alpha\beta t^{\alpha-1}$$

$$S(t|\alpha, \beta) = e^{-\beta t^\alpha}$$

In the Weibull distribution, the hazard rate is decreasing for $\alpha$ values below one, increasing for $\alpha$ values above one, and constant for $\alpha$ values equal to or greater than one [1]. When the scale parameter is equal to 4 and the shape parameters have different values, the Weibull probability distribution function changes, as shown in Fig 1. It is obvious that raising the shape parameter causes the survival curve to shift to the right. The data distribution is bell-shaped and right-skewed in Fig 1 with scale = 4 and Shape = 2.

### 3-2- The Birnbaum-Saunders distribution

The BS distribution, abbreviated as BS, is a method for modeling fatigue life that was developed in 1969 [11] by two researchers with the same name. Its introduction was inspired by the need to address the vibration issue in commercial aircraft, which led to parts' rapid wear and tear. For this reason, the distribution function is also known as the fatigue life distribution function. The application of the BS distribution function, however, extends far beyond this matter and has been used in a variety of contexts, including survival analysis [22]. The BS density function (f) for times $t_i$ with shape parameter ($\alpha$) and scale parameter ($\beta$) is as follows:

$$t_i \sim BS(\gamma, \delta). \tag{2}$$

$$f_T(t|\gamma, \delta) = \frac{1}{\sqrt{2\pi t^3}}\frac{(t+\delta)}{2\gamma\sqrt{\delta}}e^{\frac{-1}{2\gamma^2}\left(\frac{t}{\delta}+\frac{\delta}{t}-2\right)}; \ t > 0.(\gamma, \delta) > 0.$$

In fact, the BS distribution can be considered a transformation of the normal distribution. This feature makes researchers interested in using this distribution [23].

$$Z = \frac{1}{\gamma}\left(\sqrt{\frac{T}{\delta}} - \sqrt{\frac{\delta}{T}}\right) \sim N(0, 1), \ \phi(z) = \frac{1}{\sqrt{2\pi}}\exp\left(-\frac{1}{2}z^2\right), \qquad z \in \mathbb{R} \tag{3}$$

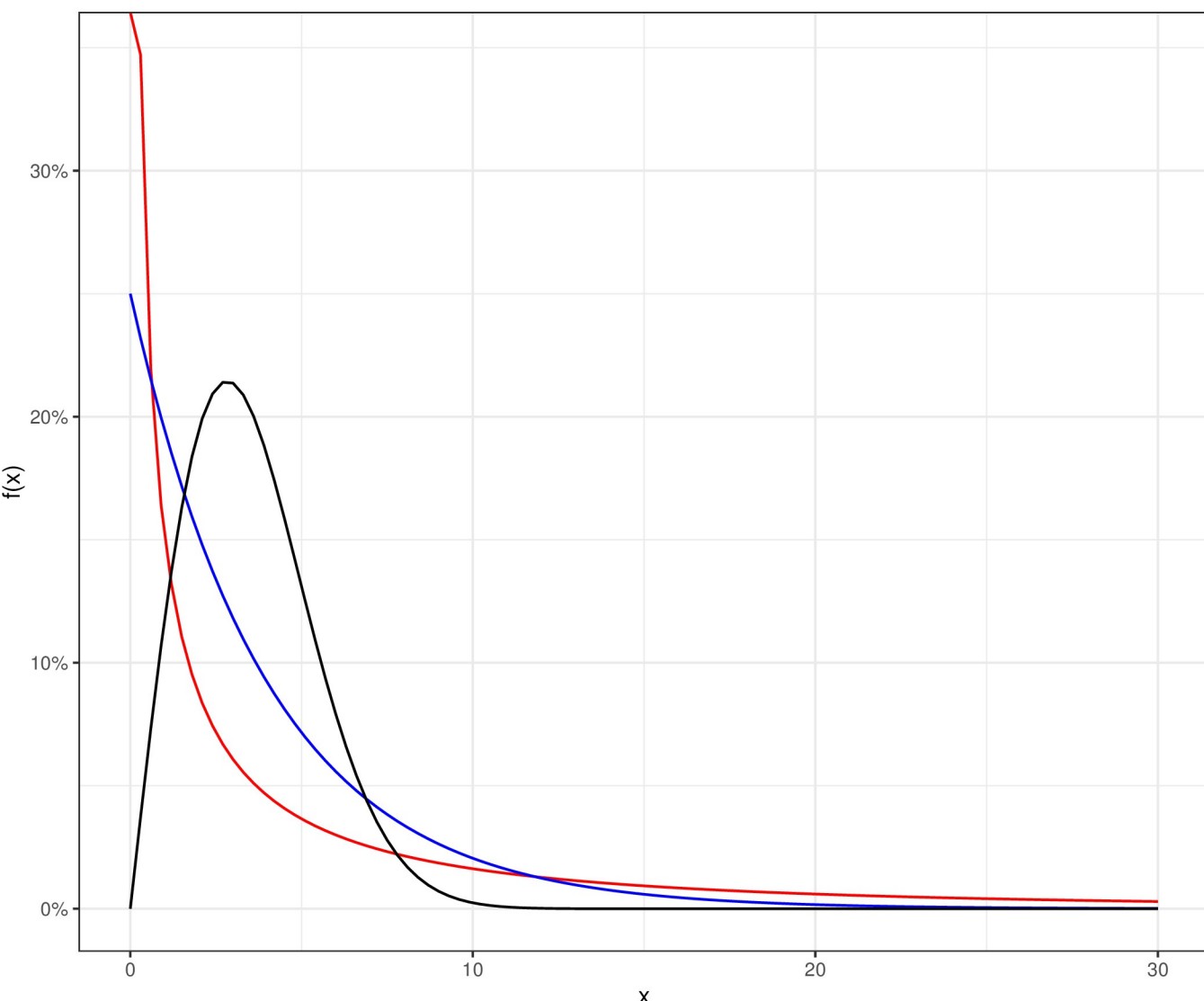

x ~ W(shape, 4), Shape: ▬ 0.5 ▬ 1 ▬ 2

**Fig 1. The Weibull probability density plot for scale parameter equal to 4 and various shapes.**

The survival function of this distribution is written as follows [23]:

$$S_T(t|\gamma, \delta) = \Phi\left(-\frac{1}{\gamma}\xi\left(\frac{t}{\delta}\right)\right) \tag{4}$$

$$\Phi(z) = \int_{-\infty}^{z} \phi(u)du, \quad \xi(u) = u^{1/2} - u^{-\frac{1}{2}} = \sinh(\log(u)), \quad u > 0$$

In Eq (4), $\Phi(z)$ is the standard normal cumulative distribution function, and $\xi(u)$ is defined. Also, $\phi(\cdot)$ is defined in Eq (1). In the BS distribution, when $\gamma$ converges to zero, the hazard rate increases. Accordingly, the increasing intensity of the hazard rate is greater for $\gamma = 0.5$ than for $\gamma = 2$ [23].

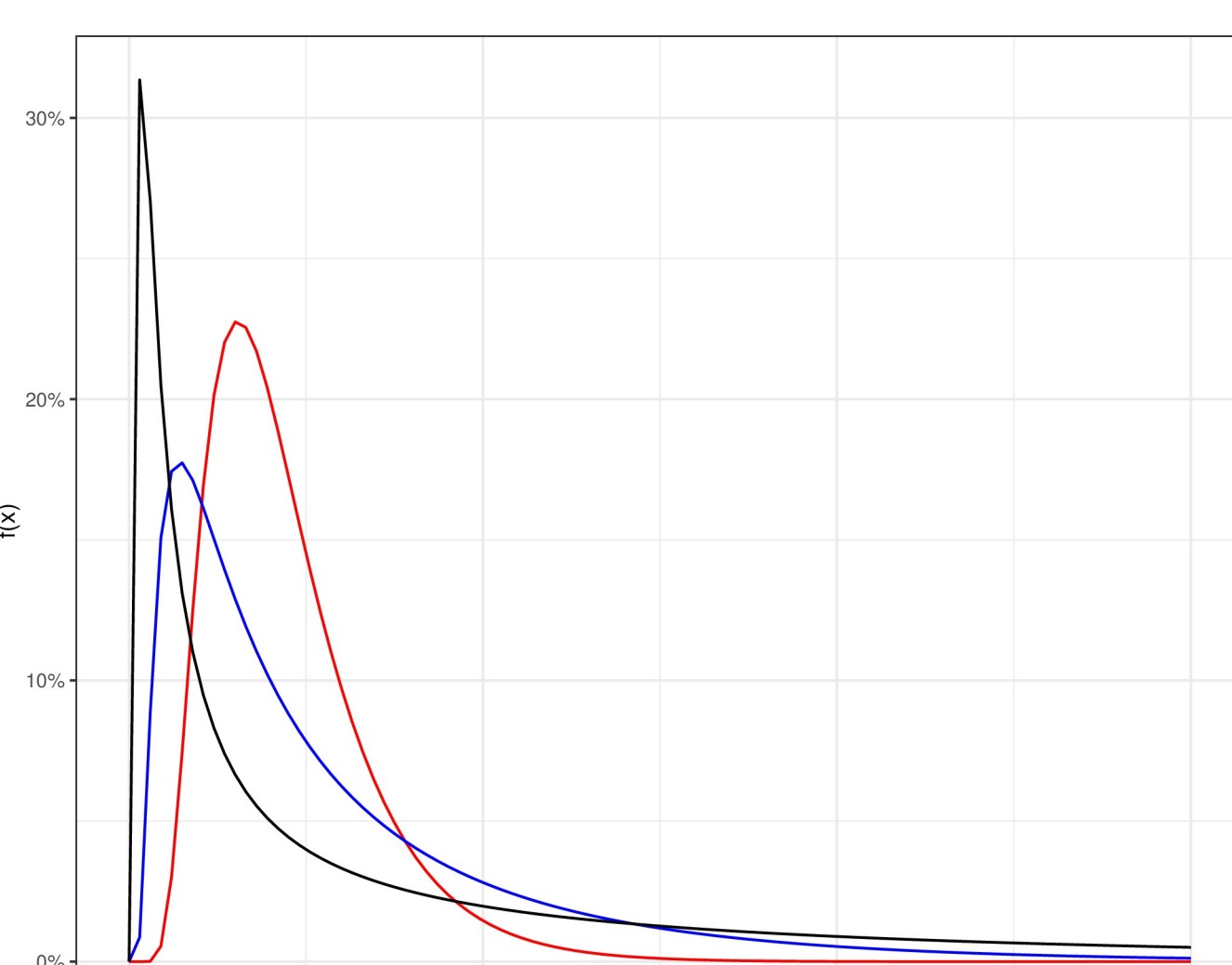

**Fig 2.** The Birnbaum-Saunders probability density plot for scale equal to 4 and various shapes.

The BS probability distribution function for the scale parameter ($\beta$) equal to 4 and different values of the shape parameter ($\alpha$) change as follows (Fig 2). Similar to the Weibull distribution, as the size of the shape parameter increases, the curve shifts to the right (increasing survival). In the BS distribution, similar to the Weibull distribution, under scale = 4 and shape = 1, the data distribution is bell-shaped and right-skewed. Curves have a low probability.

## 4- The Parametric Bayesian Approach

The BA is always recommended as an alternative to the frequentist approach. Although the benefits listed for the BA are suitable and valuable, using the BA has its difficulties. Survival data have censoring, as was mentioned in the Section 3, and fitting survival models in the presence of censored data using a frequentist approach is challenging, especially when the

censoring has a complex pattern. However, with the BA, this is made possible via Gibbs sampling techniques, Markov Chain Monte Carlo (MCMC), and special BA software (BUGS). Additionally, the Bayesian technique is adaptable to various sample sizes and allows for accurate inferences to be drawn even from small samples. This is especially helpful when estimating variance and looking at survival data. Since variance estimation is often asymptotic when using the frequentist method. The prior distribution is a highly useful piece of information that we have access to thanks to the BA. When a non-informative prior distribution is used, the BA becomes equivalent to the frequentist approach. As a result, you can apply the frequentist technique by adopting the non-informative prior distribution if you are unsure about the prior distribution of the parameters [8].

Although the advantages mentioned for the BA are appropriate and valuable, the use of the BA also has its complications. Choosing the appropriate prior distribution is one of these cases [24]. In the BA, you can choose a prior distribution that is conjugate to the likelihood function. In this case, the prior and posterior distributions are from the same family, and the estimation of posterior distribution parameters becomes easy. However, conjugate distributions cannot always be used.

Whenever no prior information is available, it is impossible to choose the prior distribution based on a subjective basis, and the conjugate prior cannot be determined. In these cases, a non-informative prior distribution is used. Laplace and Jeffries prior distributions are such distributions [25]. Solving the posterior function to determine the parameters is a further challenge for the BA. When the prior distribution is not conjugate, we may have a posterior function that doesn't have a closed form and can't be solved easily. In these circumstances, the posterior function can be solved numerically using simulation techniques, particularly Gibbs sampling. The application of the BA has been made easier over the past few decades as a result of the advancement of Bayesian methods, particularly the MCMC algorithm and the BUGS and JAGS software. This is especially true for posterior functions that lack closed form [26].

## 4-1- Density function, mean, and conditional median in a Parametric Bayesian Approach

The BA includes computational elements like the likelihood function and prior distribution, and censoring is a common feature of survival research. However, the BA uses the information provided by the censored data in the likelihood function portion, and the censoring is not taken into account while choosing the prior distribution. To impute censored values using the BA, we suppose that the observed times $t = (t_1,\ldots,t_n)$ have a density function $f(t|\theta)$ with parameter $\theta$ and that the aforementioned parameter also has a prior distribution $p(\theta)$. According to Bayes' theorem, using the equation $p(\theta|D) \propto p(D|\theta)p(\theta)$, the prior distribution is updated to the posterior distribution using the information in the data (D), the prior distribution is updated to the posterior distribution in accordance with Bayes' theorem using the equation $(p(\theta|D) \propto p(D|\theta)p(\theta))$. In the survival analysis, the real-time of the censored is greater than the recorded time (right censor). Based on this, three important indicators are stated. First, I write the conditional density function in the form of $f(t|T{\geq}c,\theta)$ which expresses the density of survival times under the condition of right-censored times (c) Within the context of the survival analysis, the right censor indicates that the actual time spent by the censor is longer than the time that was recorded. Based on this, three important signs are given. First, I formulate the conditional density function as $f(t|T{\geq}c,\theta)$, which expresses the survival time density under the condition of right-censored times (c).

$$f(t|T \geq c, \theta) = \frac{f(t|\theta)}{S(c|\theta)} \tag{5}$$

In this equation, $S(c|\theta)$ is the survival function of censoring times and is defined as $S(c|\theta) = P(T \geq c|\theta)$. Then the conditional mean and median ($S(T|T \geq c, \theta) = 0.5$)) are written as follows:

$$E(T|T \geq c) = \frac{\int_c^\infty t f(t|\theta) dt}{S(c|\theta)} \qquad (6)$$

$$0.5 = S(T|T \geq c, \theta) = \frac{P(T \geq t_{med}|\theta)}{P(T \geq c|\theta)} = \frac{S(t_{med}|\theta)}{S(c|\theta)}$$

$$t_{med} = S^{-1}(0.5*S(c|\theta)).$$

In the imputation section of the study, when generating samples, we use the proposed conditional distribution function under the condition of the censoring time observed below.

$$f(t|T \geq c, D) = \int f(t|T \geq c, \theta) p(\theta|D) d\theta \qquad (7)$$

Now we assume that we have the posterior distribution $p(\theta|D)$ and that we can produce a random sample from this distribution as $\theta^{(1)}, \theta^{(2)} \ldots, \theta^{(s)}$, where $\theta^{(s)}$ is equal to the s-th member of this set. Using the mentioned samples and the conditional distribution function $f(t|T \geq c, \theta^{(k)})$, we can produce $t^{(k)}$ samples. We generate samples in the same way as a Monte Carlo sample. We can estimate the mean and median of the proposed conditional distribution by finding the mean and median of the samples. By simulating the censored times and substituting them for the indicated periods in the original data using MCMC methods [23], we can obtain the whole data set, which includes both the actual observed times and the imputed times. Using common graphic techniques, we can evaluate the effectiveness of simulation and imputation.

## 4-2- Application of the Weibull failure time distribution in Parametric Bayesian Approach

In this section, using the topics of Section 3, we intend to examine the use of the Weibull distribution in the BA. One of the complications of using the Weibull distribution in the BA is determining the prior distribution function for its parameters. It should be noted that whenever the shape and scale parameters of this distribution are unknown, there is no prior conjugate distribution function for this distribution [21]. We presume that the Weibull distribution's shape and scale parameters have a gamma prior distribution function in this study [21, 25].

$$t_i \sim Weibull(\alpha, \beta); \ \alpha \sim Gamma(\alpha_1, \beta_1); \ \beta \sim Gamma(\alpha_2, \beta_2) \qquad (8)$$

In the above relation, the set $\{\alpha_1.\alpha_2.\beta_1.\beta_2\}$ is the hyperparameters of the Weibull distribution. In this case, the posterior distribution does not have a closed form, and simulation of the suggested values of censored observations is done by MCMC methods on the condition of observed censoring times. For this purpose, it is necessary to write conditional Eqs (5–7) for

the Weibull distribution.

$$f(t|T \geq c, \alpha, \beta) = \frac{\alpha \beta t^{\alpha-1} e^{-\beta t^\alpha}}{e^{-\beta c^\alpha}} \tag{9}$$

$$E(T|T \geq c) = \frac{\int_c^\infty t \times \alpha \beta t^{\alpha-1} e^{-\beta t^\alpha} dt}{S(c|\alpha, \beta)} = \frac{1}{S(c|\alpha, \beta)} \left\{ [-te^{-\beta t^\alpha}]_c^\infty - \int_c^\infty -e^{-\beta t^\alpha} dt \right\} \tag{10}$$

$$= c + \frac{1}{e^{-\beta c^\alpha}} \int_c^\infty e^{-\beta t^\alpha} dt$$

If we take the Weibull distribution with parameter $\alpha = 1$ (an exponential distribution with parameter β), then the conditional mean (10) can be condensed to the following form, which also includes the unconditional mean.

$$E(T|T \geq c, \alpha, \beta) = c + \frac{1}{e^{-\beta c}} \int_c^\infty e^{-\beta t} dt = c + \frac{1}{\beta} = c + E(T) \tag{11}$$

But under $\alpha > 1$, the integral (10) does not have a closed answer, and we use numerical methods to calculate the conditional average. To find the conditional median, we use the conditional survival function under the condition of censoring. For this purpose, we have:

$$S(T|T \geq c) = \frac{P(T \geq t_{med})}{P(T \geq c)} = 0.5$$

$$\Rightarrow \frac{S(t_{med})}{S(c)} = \frac{e^{-\beta t_{med}^\alpha}}{e^{-\beta c^\alpha}} = 0.5 \Rightarrow e^{-\beta(t_{med}^\alpha - c^\alpha)} = 0.5 \Rightarrow t_{med}^\alpha - c^\alpha = \frac{-\ln(0.5)}{\beta}$$

$$t_{med} = \sqrt{\frac{\ln(2)}{\beta} + c^\alpha} \tag{12}$$

By knowing the censoring time and the size of the parameters in Eq (12), the median value can be determined for each censored observation. We generate random samples from Eq (7) and the MCMC method under the condition of censored observations.

## 4-3- Application of the Birnbaum-Saunders distribution (fatigue life) in a Parametric Bayesian Approach

In this Section, using the topics of Section 3, we intend to examine the application of the BS distribution in the BA. The prior distribution for shape (γ) and scale (δ) parameters is as follows (18).

$$t_i \sim BS(\gamma, \delta), \ \gamma \sim Gamma(\gamma_1, \gamma_2), \ \delta \sim Gamma(\delta_1, \delta_2) \tag{13}$$

The conditional mean of the variable $T$ is written as follows under the condition of

censoring the observations:

$$E(T|T \geq c, \gamma, \delta) = \frac{\int_c^\infty t*f(t|\gamma, \delta)dt}{S(c|\gamma, \delta)} \qquad (14)$$

$$= \frac{\int_c^\infty \frac{t}{\sqrt{2\pi t^3}} \frac{(t+\delta)}{2\gamma\sqrt{\delta}} e^{\frac{-1}{2\gamma^2}\left(\frac{t}{\delta}+\frac{\delta}{t}-2\right)}dt}{\Phi\left(-\frac{1}{\gamma}\xi\left(\frac{c}{\delta}\right)\right)} = \frac{\int_c^\infty \frac{1}{\sqrt{2\pi t}} \frac{(t+\delta)}{2\gamma\sqrt{\delta}} e^{\frac{-1}{2\gamma^2}\left(\frac{t}{\delta}+\frac{\delta}{t}-2\right)}dt}{\int_{-\infty}^{\frac{-1}{\gamma}\xi\left(\frac{c}{\delta}\right)} \phi(z)dz}$$

The conditional mean of the random variable *T* does not have a closed form if the observations are censored. To determine the quantiles of this distribution (*q*), we use the following Eq (15).

$$t(q;\ \gamma, \delta) = F_T^{-1}(p;\ \gamma, \delta) = \delta\left(\frac{\gamma z(q)}{2} + \sqrt{\left(\frac{\gamma z(q)}{2}\right)^2 + 1}\right)^2 \qquad (15)$$

According to Eq (15), $t(0.5;\ \gamma, \delta) = \delta$. Therefore, in the BS distribution, the median of the distribution is equal to the scale parameter ($\delta$) [23]. Considering that the survival times are often skewed, it is suggested to use the median of the simulated times compared to their average. In the BS distribution, the median of the simulated times is equal to the scale parameter, which facilitates the calculations.

## 4-4- Fix of censoring in Bayesian Approach

To conduct a performance comparison between the parametric BA using the Weibull distribution and the BS distribution, it is necessary to establish fixed censoring sizes (*p*) at nominal values of 0.10, 0.20, and 0.50. In this study, censored data are produced from the exponential distribution using the parameter $\theta$, and the magnitude of this parameter should be computed so that the value of p is equal to the percentage of censoring. To achieve the required censoring, the selection of the exponential distribution parameter is contingent upon identifying the magnitude of the shape parameter in either the Weibull distribution or the BS distribution. This study assumes that the censoring mechanism follows the Missing At Random (MAR) assumption.

## 5- Data simulation process

The data simulation will have sample sizes of 100, 200, and 300. We follow the steps below to simulate the data in R software for a sample size of 200 with 10% censoring. We do the same for other sample sizes.

1. The first step is to determine the values of the distribution parameters. We will use the Weibull, or BS distribution in the simulation of the observed times. Both distributions have two parameters. The shape parameter of each fixed distribution is determined to be equal to 0.5, 1, or 2 in each run. The scale parameter of each distribution is determined as a regression equation: *Scale* = exp ($b_1 + b_2 \times x$), where variable x is a two-state variable with probabilities of 0.50 and 0.50 and the values of the regression coefficients are chosen so that the average values of the scale parameter are simulated to be near to the desired value (here equal to 4). Due to the different probability density functions of the Weibull distribution in R (Proportional Hazard (PH) model) and BUGS (accelerated failure time (AFT) model), $\beta^{\left(-\frac{1}{2}\right)}$ transformation has been used for BUGS. This has caused the regression coefficients to differ in

the Weibull distribution, while the size of the regression coefficients is similar for the BS distribution.

2. From step one, we have determined the parameters of the Weibull, or BS distribution. Now we can generate 200 data points from the desired distribution randomly.

3. When the size of the Weibull distribution parameter is selected, we can choose its value by changing the size of the exponential distribution parameter so that the desired values for censoring the data (p = 10, 20, or 50%) are obtained.

4. By using the parameter size of the exponential distribution obtained from step 3, the random generation of censoring times is done with 200 data points.

5. Combining observed and censored times: We combine the observed and censored times to have a dataset with the desired percentage of censoring (initial data).

6. Using the R and R2openBUGS software packages, the censored values in the primary data are simulated 11,000 times (thinning = 10), where the burn-in process is performed 1000 times, and 10,000 final simulated points will remain for each censoring time.

7. From step 6 and for each censoring time, we will have 10,000 simulated times. The mean and median have been used instead of simulated times.

8. The augmented data includes the observed times and the imputed mean (or median) for the censors. The simulation results are reported in the original article with a sample size of 200 and in the appendix of the article with sample sizes of 100 and 300.

9. In the simulation section, curves are produced for each of the 27 scenarios involving the Weibull and Birnbaum-Saunders distributions. These scenarios include three sample sizes (100, 200, and 300), three censoring percentages (10, 20, and 50%), and three different shape parameters (0.5, 1, and 2). The Geweke diagnostic is used to evaluate convergence in parameter simulation; autocorrelation in simulation values is represented by an ACF plot; and a trace plot is used to express the trace of parameter generation. The next step is to verify the independent sample generation status using the Effective Sample Size (ESS). Ultimately, a drawing of the posterior density diagram representing the distributions' parameters will be made. It should be pointed out that the Geweke diagnostic is deemed acceptable in the range (-2, +2) [27], and for most applications, the ESS index is suitable for values over 1000 samples [28, 29].

10. Similar to Section 9, graphs and indicators pertaining to the simulation of censoring times in cancer data under two the Weibull and Birnbaum-Saunders distributions will be presented. The Deviance Information Criteria (DIC) are the final criterion for selecting one of the distributions.

## 5-1- Evaluating the suitability of simulations

In sections 5–2 to 5–4, censored times in breast cancer data and censored times under the Weibull and BS distribution simulations will be provided. Before proceeding with the aforementioned sections, it is necessary to verify the suitability of the simulations. Fig 3 depicts the autocorrelation for the posterior simulation of the Shape, b1, and b2 parameters of the Weibull and BS distributions. Note that the shape, scale, and censoring parameters of Fig 3 are set to 2, 4, and 20%, respectively. Furthermore, Fig 3 reports the amount of autocorrelation for the two parameters, shape and scale (the Weibull and BS distributions), in the context of breast cancer data. Furthermore, Fig 3 reports the amount of autocorrelation for the two parameters, shape

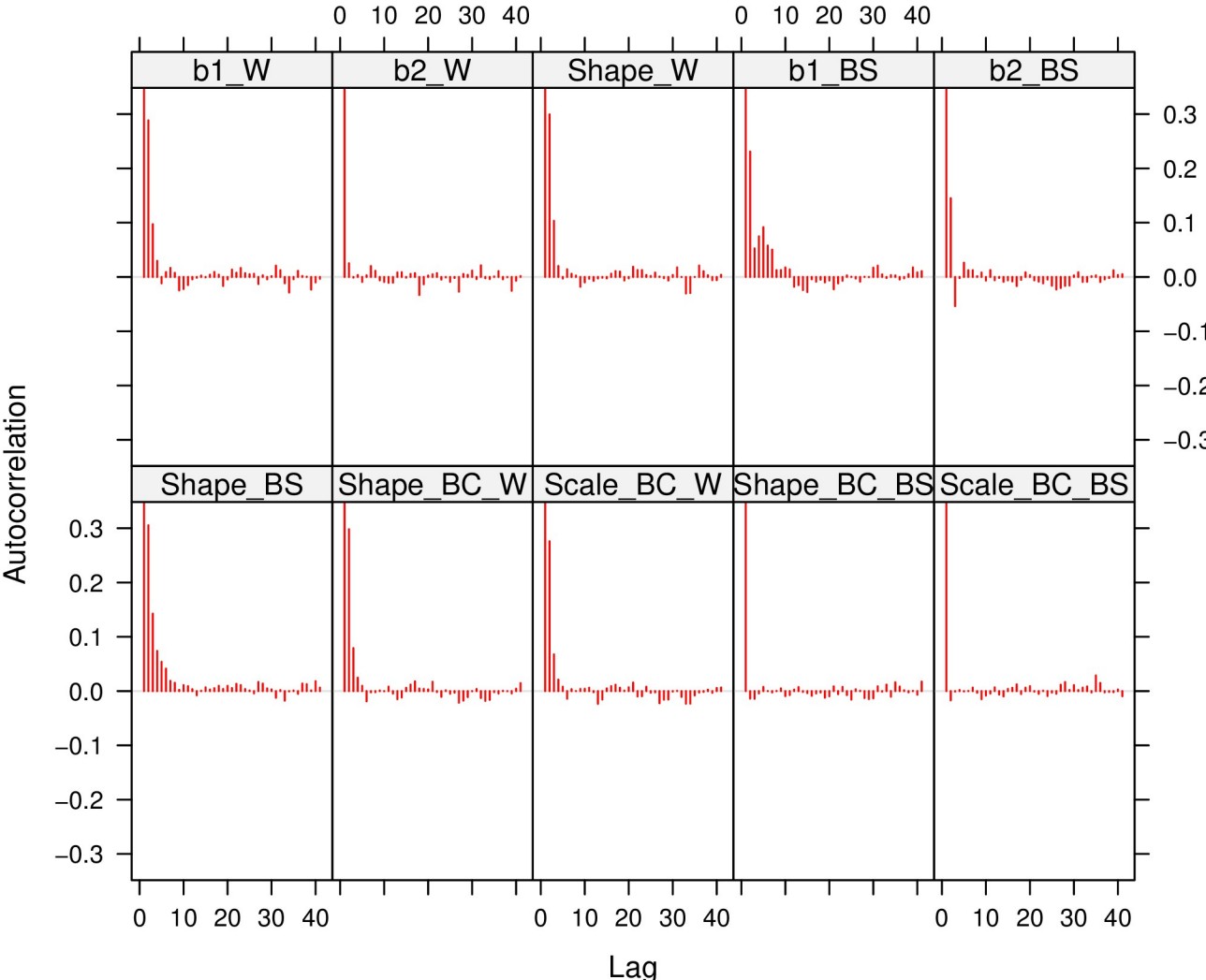

**Fig 3. Autocorrelation for different data simulation scenarios, include: The Weibull (b1_W, b2_W, Shape_W), the Birnbaum-Sanders (b1_BS, b2_BS, Shape_BS) distribution parameters, and the BC scenario parameters (the Weibull: Shape_BC_W, Scale_BC_W, the BS: Shape_BC_BS, Scale_BC_BS).**

and scale (the Weibull and BS distributions), in the context of breast cancer data. Figs 4 and 5 depict the posterior distribution and trace plot of the parameters indicated in this scenario, respectively. Table 1 shows the amount of the ESS index for similar scenarios.

To verify that the simulated parameter sizes were converging, we employed the Geweke diagnostic. Fig 6 illustrates this index for a range of sample sizes (100, 200, and 300), censoring percentages (10, 20, and 30%), and shape parameters (0.5, 1, and 2). It should be noted that the scenario's parameter values are written in brackets in Fig 4.

The convergence of parameter values under the Weibull (Shape_Weibull, Scale_Weibull) and BS (Shape_BS, Scale_BS) distributions for breast cancer data is displayed in Fig 7.

## 5-2- Examination of simulated data curves from the Weibull distribution

Similar to what was described in Section 4, the observed times are from the Weibull distribution, and the censors are simulated from the exponential distribution. The initial data is made

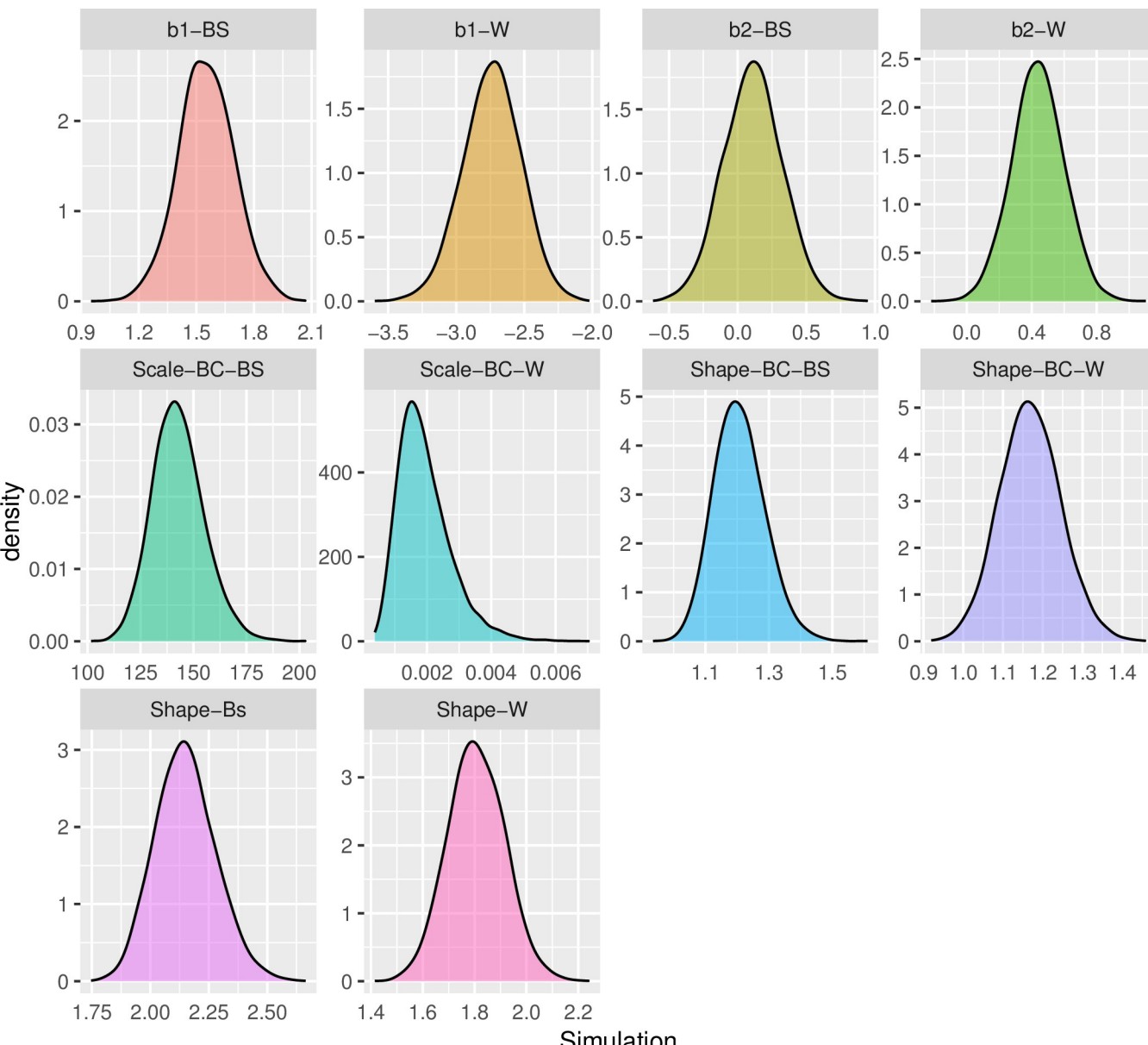

**Fig 4. Posterior distributions for different data simulation scenarios, include: The Weibull (b1_W, b2_W, Shape_W), the Birnbaum-Sanders (b1_BS, b2_BS, Shape_BS) distribution parameters, and the BC scenario parameters (the Weibull: Shape_BC_W, Scale_BC_W, the BS: Shape_BC_BS, Scale_BC_BS).**

according to paragraphs one to five, and this data is used to draw the Kaplan-Meier curve. Also, the times observed in the initial data have been used to draw omitting censored curves. The Weibull distribution curves are drawn from the completed data described in paragraph 8 of Section 4. The curves for the Weibull distribution (N = 200) with three different values of the shape parameter and a constant value of the scale parameter equal to 4 are drawn in this section, and other curves (N = 100 and 300) are presented in the appendix. It should be noted that the choice of shape and scale parameter values in distributions is optional. Due to the different shapes of the distribution for values less than and greater than one for the shape parameter, its value has been chosen as 0.5, 1, and 2. The exponential distribution parameter ($\theta$) size

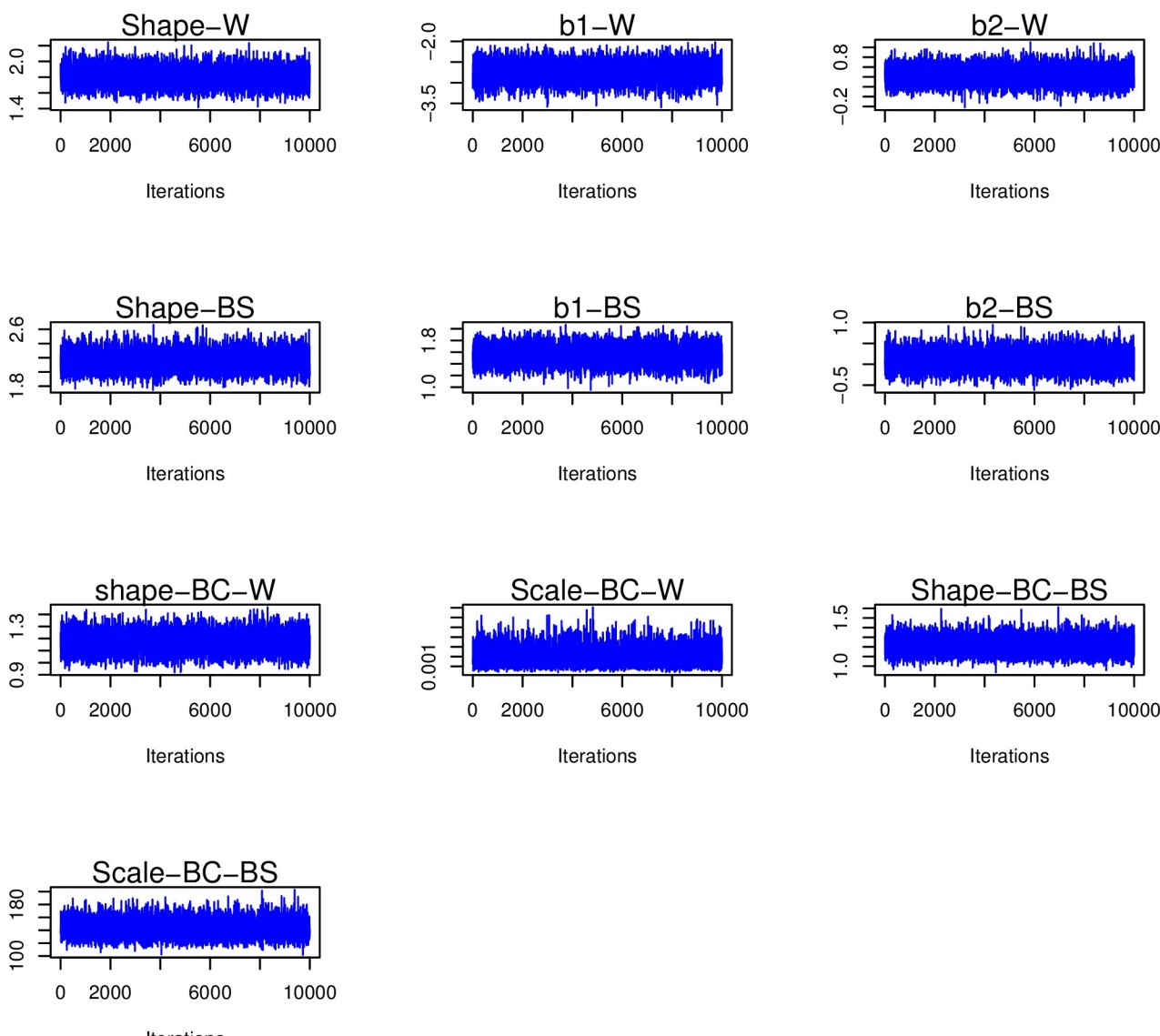

**Fig 5. Trace plots for different data simulation scenarios include: The Weibull (b1_W, b2_W, Shape_W), the Birnbaum-Sanders (b1_BS, b2_BS, Shape_BS) distribution parameters, and BC scenario parameters (the Weibull: Shape_BC_W, Scale_BC_W, the BS: Shape_BC_BS, Scale_BC_BS).**

for various the Weibull distribution shape ($\alpha$) parameter values and censorship percentages ($p$) is presented in Table 2. In section 4, paragraph 5 provides the formula for calculating the parameter's size.

**Table 1. The ESS index for the Weibull and BS distributions in simulation and the BC dataset scenarios.**

| Distribution | Sample Size | Censoring Percent | Parameter | | | |
|---|---|---|---|---|---|---|
| | | | Shape | b1 | b2 | Scale |
| Simulation: Weibull | 200 | 20% | 5259.25 | 5642.21 | 9507.72 | - |
| Simulation: BS | 200 | 20% | 4287.13 | 4373.32 | 5217.23 | - |
| BC data: Weibull | 220 | 40% | 5405.83 | - | - | 5673.36 |
| BC data: BS | 220 | 40% | 8680.22 | - | - | 9202.25 |

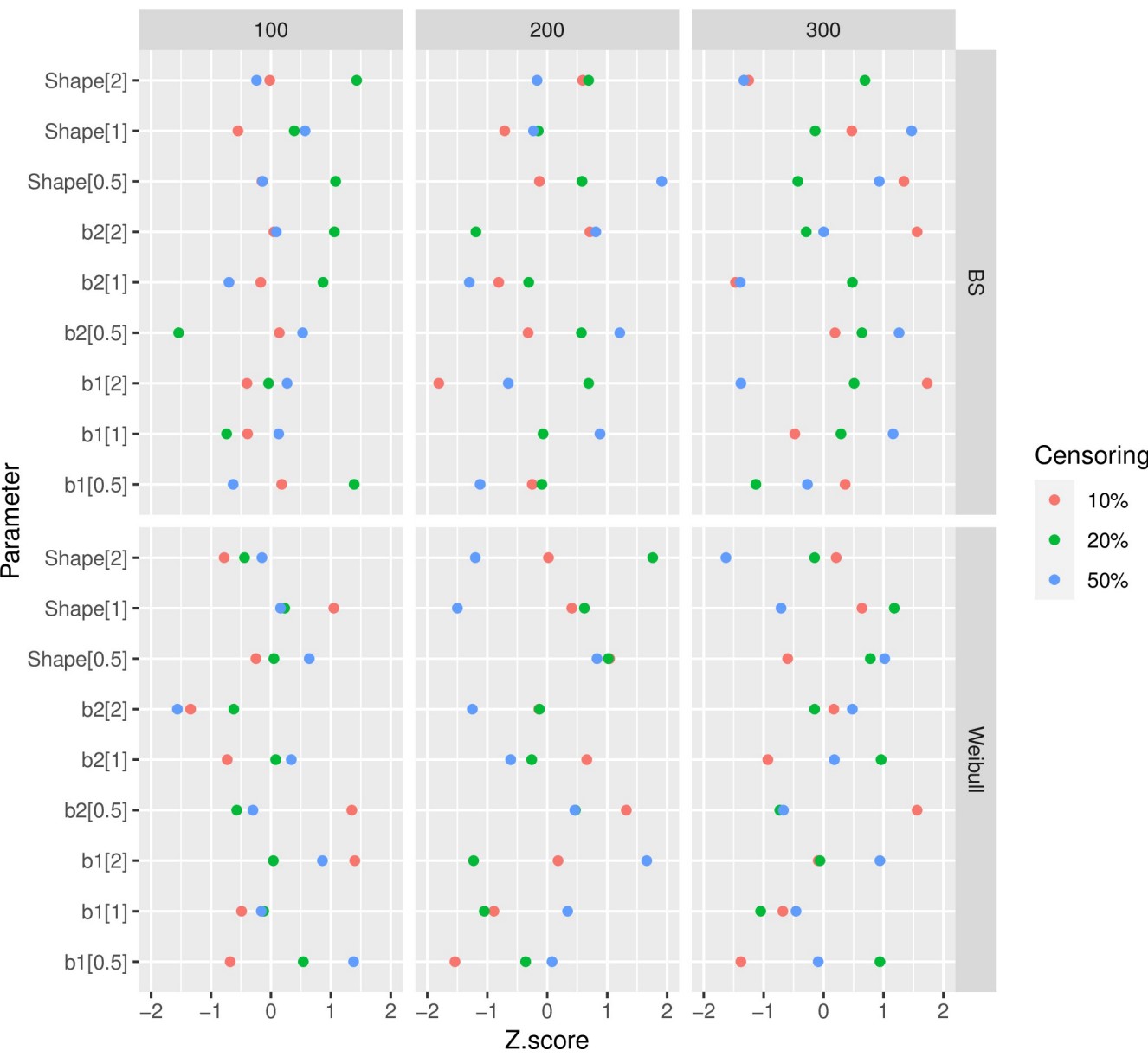

**Fig 6. Convergence for different data simulation scenarios under the Weibull and Birnbaum-Saunders distributions, which are adjusted according to the sample size (100, 200, and 300) and the censoring percentage.**

The size of the regression coefficients of the scale parameter of the Weibull distribution is reported in Table 3 for parameters 0.5, 1 and 2, respectively. The values of the regression coefficients have been chosen so that the scale parameter is equal to 4.

In the graphs of Fig 8, Kaplan-Meier curves, BA with the median of simulated times, and omitting censored curves (censored data are removed) are reported for a sample size of 200. To see the curves for sample sizes of 100 and 300, review the supplementary files (S1 and S2 Figs).

Under constant sample size and censoring percentage, it is observed that the shape of the curve moves to the right (survival improves) as the parameter size increases. According to the nature of censoring and its function in calculating the survival rate, under the constant shape

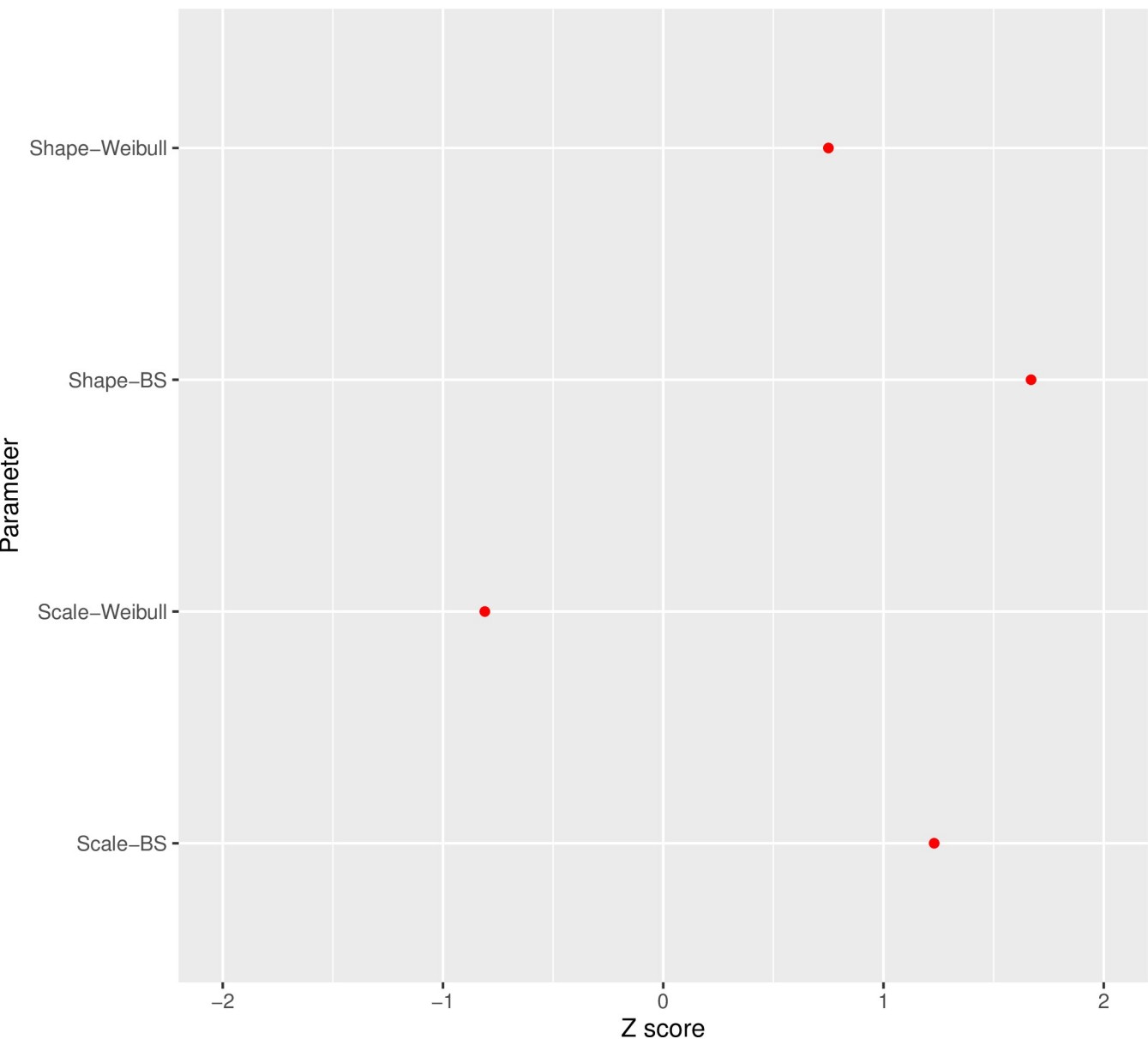

**Fig 7.** Convergence for different BC data simulation scenarios under the Weibull and Birnbaum-Saunders distributions.

parameter and sample size, the increase in censoring moves the curve horizontally. Also, with the increase in the percentage of censoring and under the parameters of shape and fixed sample size, the BA curve takes on a steeper downward slope.

**Table 2. The values of the parameter $\theta$ of the exponential distribution for various percentages of censoring and various values of parameters of the Weibull distribution.**

| Censoring Percent / Weibull Parameter | 0.10 | 0.20 | 0.50 |
|---|---|---|---|
| W (0.5, 4) | 0.02 | 0.04 | 0.30 |
| W (1, 4) | 0.04 | 0.08 | 0.30 |
| W (2, 4) | 0.02 | 0.06 | 0.20 |

**Table 3. The values of regression coefficients for different values of the Weibull distribution shape parameter.**

| Shape | Coefficient | $b_1$ | $b_2$ |
|---|---|---|---|
| 0.5 | | -0.80 | 0.20 |
| 1 | | -1.5 | 0.40 |
| 2 | | -3 | 0.3 |

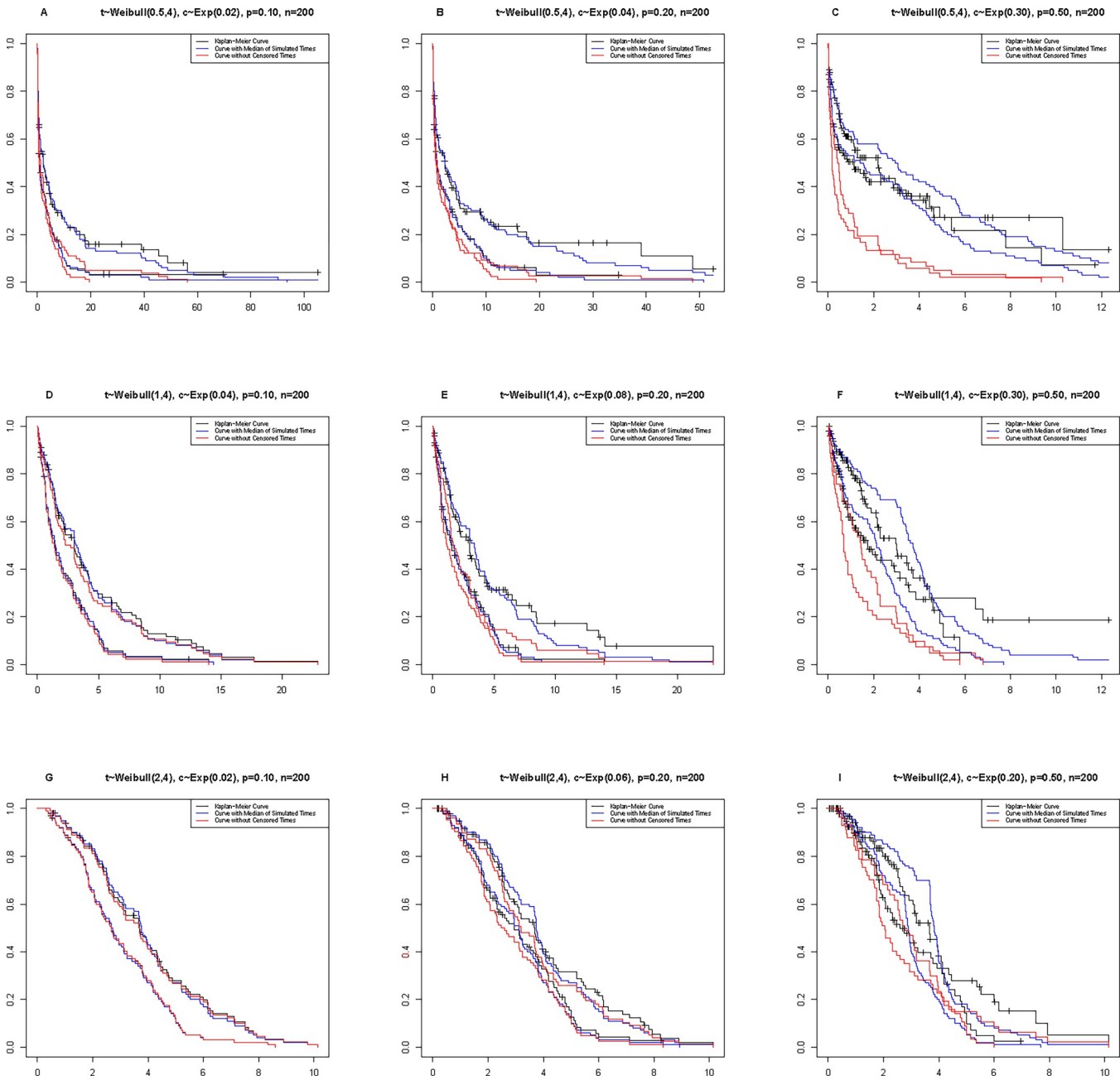

**Fig 8. Kaplan-Meier curves, BA (median of simulated times), and observed times (omitting censored) for each of the scenarios listed in Table 1.**

The curves in Fig 9 show that the range of simulation changes for censored times increases with the increase in censoring percentage.

Fig 9 indicates that the average of imputations is a good representative of the simulated values. It is clear that the range of changes in simulated numbers grows as the percentage of censoring goes up. This is especially true with censoring by 50%. In general, by comparing the curves related to the median and the mean under the shape parameter, censoring percentage, and the same sample size, it can be concluded that both the median and mean indicators can be considered suitable alternatives for simulations.

## 5-3- Examination of simulated data curves from the Birnbaum-Saunders distribution

The steps in this part are similar to those in Section 5–1. The exponential distribution will be used for the censored times and the BS distribution for the observed times. The research on the analysis of survival data has increasingly utilized the BS distribution in recent years [30–33]. In this study, we will compare the performance of this distribution in the survival analysis. Similar to the Weibull distribution, curves are drawn based on the details in Tables 4 and 5.

In Fig 10, the curves are similar to those in Fig 8, but have been drawn under the BS distribution. Under the fixed sample size and censoring percentage, the shape of the survival curve moves to the right as the parameter size increases (survival increases). According to the nature of censoring and its function in calculating the survival rate, under the constant shape parameter and sample size, the increase in censoring causes horizontal movement of the curve. Also, with the increase in the percentage of censoring and under the parameters of shape and fixed sample size, the BA curve takes on a steeper downward slope. In fact, the behavior of the BA in the imputation of censors under the BS distribution is similar to that of the Weibull.

Fig 11 shows how the average censoring times change over time. When the censoring level is set to 50%, the average simulation times in the middle of the curve drop quickly. Under 50% censored data, it appears that the BS distribution has limited ability to simulate the censored data, at least for a part of the data.

## 5-4- Investigating the performance of the Weibull and the Birnbaum-Saunders distributions in real breast cancer data

In this part, the effectiveness of the BA is examined in the simulation of censoring times for the BC data from the Cancer Research Center (CRC) at Shahid Beheshti University of Medical Sciences. The BC data, which contains 220 patients who were identified and followed up between 2015 and 2023, was made accessible on February 1st, 2023. The sample size in this study is 220 women because only patients with cancer recurrences were included. The present article was conducted and approved as a research project by the Ethical Committee in biomedical research with the code IR.MODARES.REC.1401.171. The time to the event or censoring is related to the time interval between admission to the CRC and death or censoring. The mean (standard deviation) time to the event and censoring are, respectively, 125.27 (107.73) and 157.09 (146.09) days. Approximately 60% of the patients, or 132 individuals, have died, and 40% are censored. The patients have an average age distribution of 40 years, and based on this information, the age variable was split into two subgroups: those with an age of less than 40 years (66 individuals) and those with an age of more than 40 years (154 people). This allowed the curves to be drawn in a way that was adjusted to the age variable. Assuming that the observed times follow a the Weibull or BS distribution and that the imputation of censored times is the primary objective, 10.000 simulation times are performed for each censored time. The censoring time is then imputed using the mean or median of the simulation times. Using

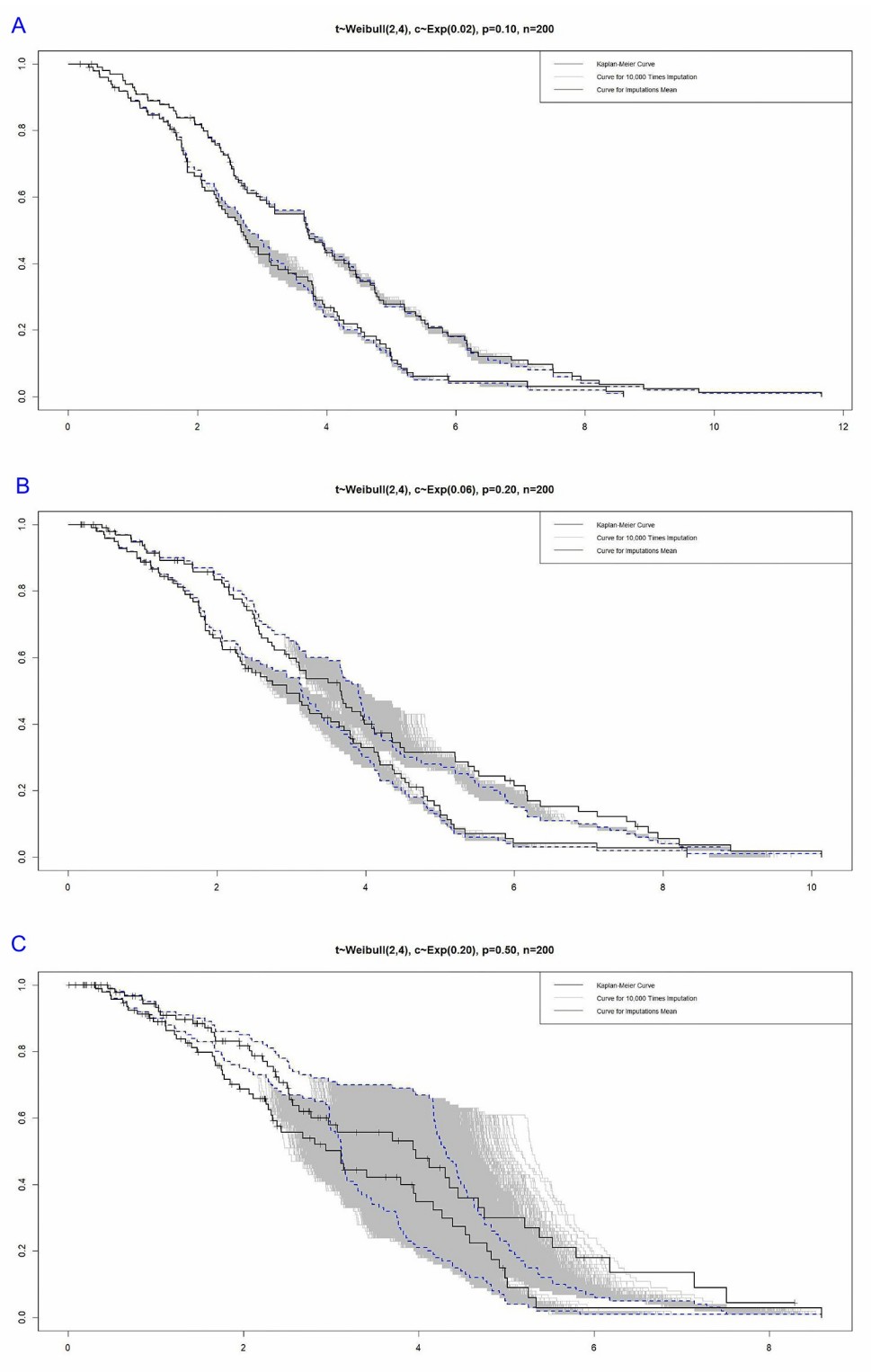

**Fig 9. Kaplan-Meier curves, 10.000 simulation times for each censoring time, along with the average simulation times.**

**Table 4. Values of the parameter θ of the exponential distribution for different percentages of censoring and different values of the BS distribution (BS) shape parameter.**

| Censoring Percent BS Parameter | 10% | | | 20% | | | 50% | | |
|---|---|---|---|---|---|---|---|---|---|
| | N = 100 | N = 200 | N = 300 | N = 100 | N = 200 | N = 300 | N = 100 | N = 200 | N = 300 |
| BS (0.5. 4) | 0.01 | 0.02 | 0.03 | 0.04 | 0.04 | 0.05 | 0.15 | 0.15 | 0.15 |
| BS (1.4) | 0.02 | 0.01 | 0.02 | 0.04 | 0.04 | 0.04 | 0.15 | 0.15 | 0.15 |
| BS (2,4) | 0.01 | 0.01 | 0.01 | 0.02 | 0.02 | 0.02 | 0.10 | 0.15 | 0.15 |

the Kaplan-Meier, BA, and failure times (omitting-censored) curves, one can determine whether or not the Weibull or BS distribution is eligible to be employed. The DIC index is also used to decide how distributions behave. It should be noted that the times in this data are expressed in days, and the curves were created by varying the participants' ages at the 40-year cutoff.

The curves in Figs 12 and 13 are drawn under the Weibull and BS distributions, respectively.

The performance of the Weibull and BS distributions is similar when we use the median of the simulated times rather than the censored times. However, the BS distribution's associated curve lowers sharply when we use their average rather than the simulated times. Only in about 50% of censoring is this condition seen. If the final selection criterion of the distribution is equal to the DIC index, the BS distribution with a DIC index size equal to 1510 and the Weibull distribution with a the DIC index size equal to 1698 perform better.

## 6- Discussion

In this study, the effectiveness of the Weibull and BS distributions in the simulation of censoring times has been compared. Two distributions' performances were compared using curves and the DIC index. There is no agreement on the validation and verification of simulated and censored data, and researchers have taken varied approaches [34]. This work has been validated and verified using the approach recommended by David et al. [35]. The validity of the study was evaluated and confirmed by using different scenarios for sample size, censoring percentage, and shape parameters. The study's verification was reviewed and confirmed by evaluating the command codes and their correct implementation in 56 scenarios. Such scenarios include 27 runs with the Weibull distribution, 27 runs with the BS distribution, and two runs with real BC data.

The suitability of the simulations is demonstrated by the graphs in Figs 3 through 7. The convergence plot (Figs 6 and 7) indicates that the simulations running under different scenarios have optimal convergence. Furthermore, autocorrelation in posterior parameters is close to zero for long lags (Fig 3), and for the majority of the parameters, the posterior distribution resembles the normal distribution (Fig 4). In addition, for 10,000 simulations, the ESS index is good for all parameters (Table 1). As a result, in general, the simulation is informative, and the simulation findings are explored more below.

**Table 5. Regression coefficients for different values of the BS distribution shape parameter.**

| Coefficient Shape | $b_1$ | $b_2$ |
|---|---|---|
| 0.5 | 1.37 | 0.15 |
| 1 | 1.37 | 0.15 |
| 2 | 1.37 | 0.15 |

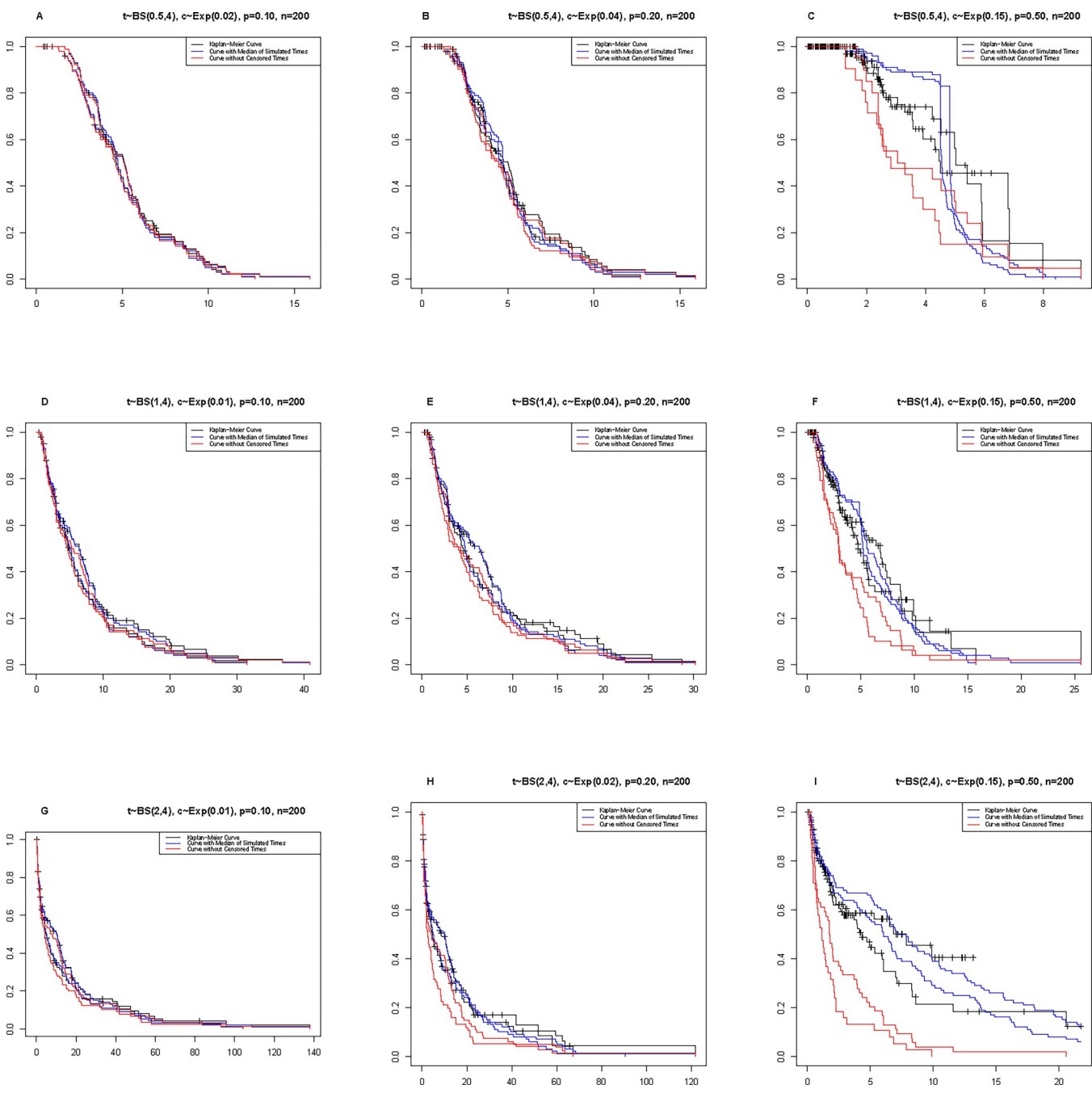

**Fig 10. Kaplan-Meier curves, the BA, and times observed from the BS distribution for each of the states listed in Table 3.**

We first check the results of the simulation. It is true that altering the values of the shape parameter, sample size, and censoring percentage would alter the appearance of the curves. For instance, as the censoring percentage rises, the effect of the imputation value on the data also rises. In both distributions, we observe that at 50% censoring, the BA's performance in data simulation is correlated with a rise in survival (the curve shifts to the right). Additionally, we looked at how the shape of the parameter affected the curves and discovered that the Weibull and BS distributions yielded different findings. In the Weibull distribution, an increase in

**A**        **t~BS(2,4), c~Exp(0.01), p=0.10, n=200**

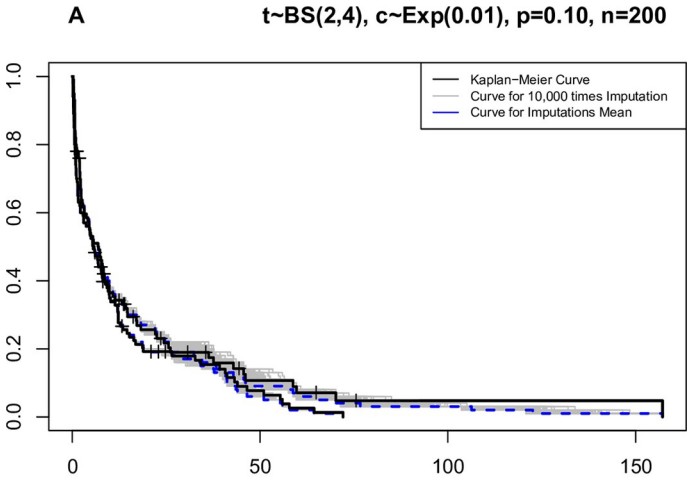

**B**        **t~BS(2,4), c~Exp(0.02), p=0.20, n=200**

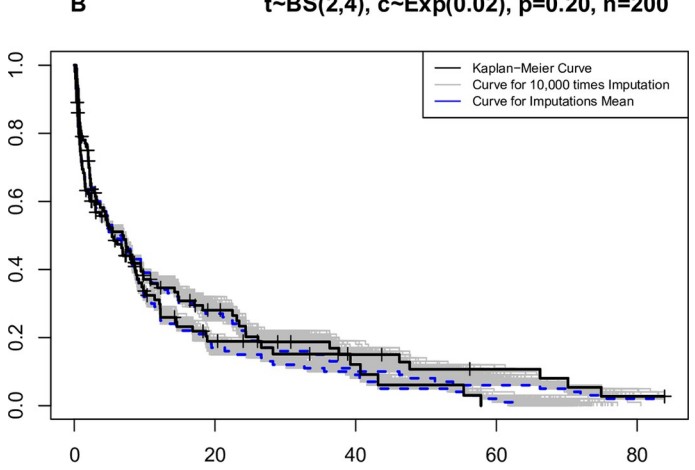

**C**        **t~BS(2,4), c~Exp(0.15), p=0.50, n=200**

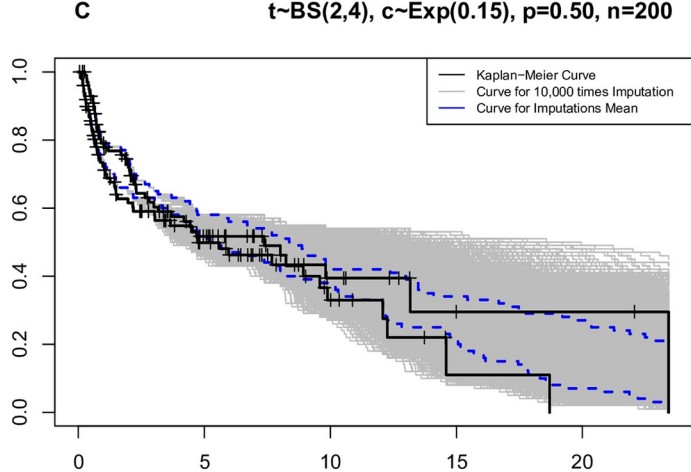

**Fig 11. Kaplan-Meier curves, 10.000 simulation times for each censoring time, along with the mean simulation times from the BS distribution.**

### Posterior Estimate: Shape=1.24,Scale=0.001,DIC=1698

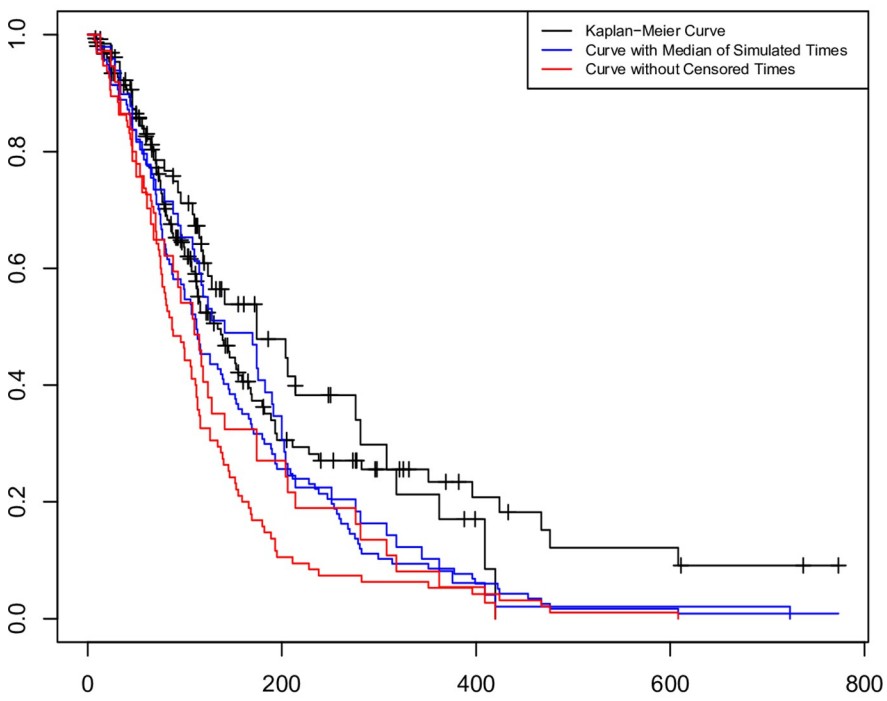

### t~Weibull, p=0.40, n=220

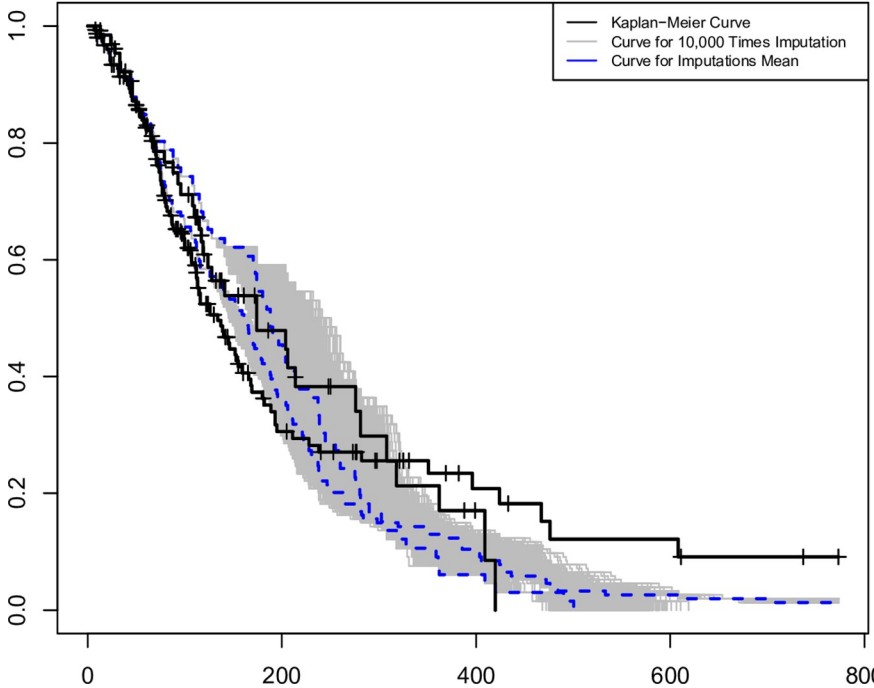

**Fig 12. Kaplan-Meier curves, observed times, mean, and median of simulated times with Weibull distribution under BA.**

## Posterior Estimate: Shape=1.22, Scale=145.21, DIC=1510

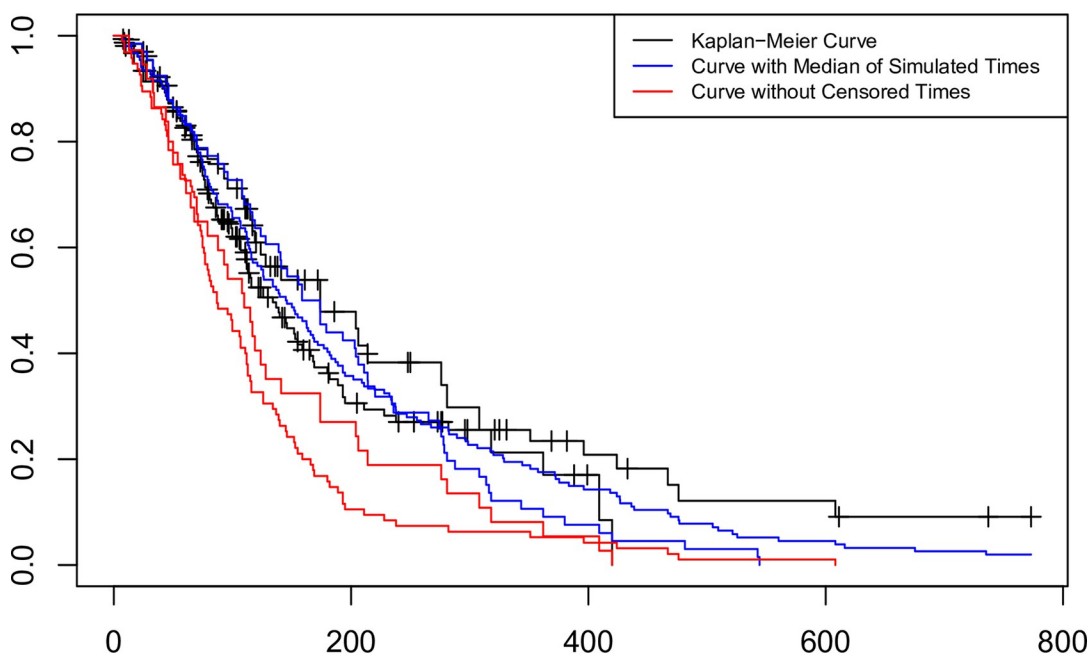

## t~Birnbaum−Saunders, p=0.40, n=220

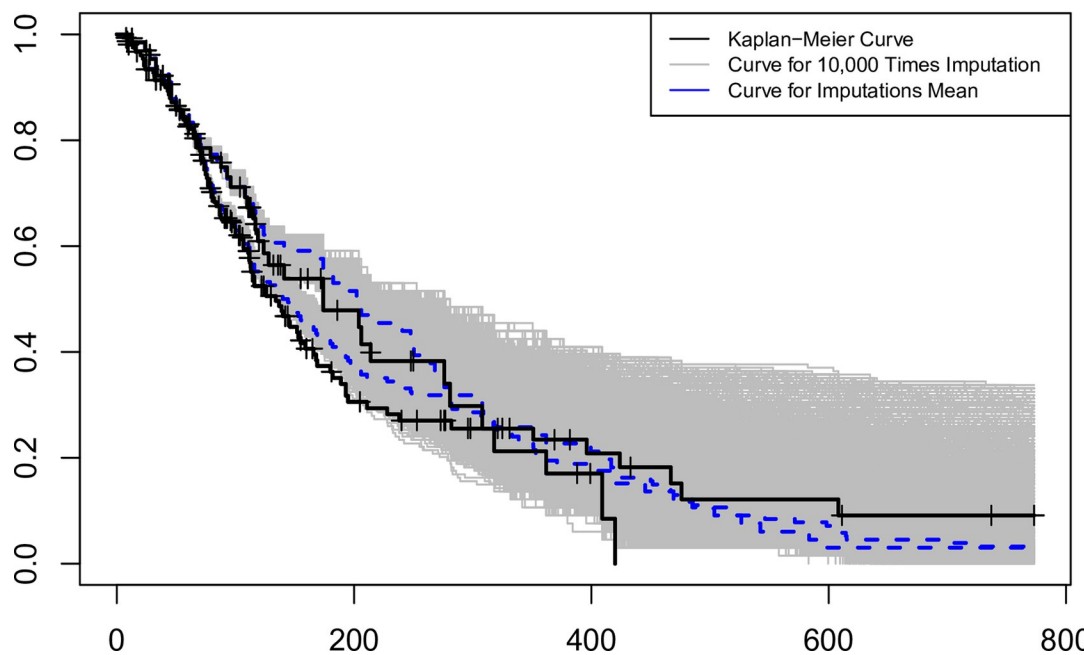

**Fig 13. Kaplan-Meier curves, observed times, mean and median of simulated times with BS distribution under BA.**

the shape parameter results in an increase in survival, but in the BS distribution, an increase in the shape parameter results in a drop in survival. Finally, the matching curves were generated to allow the user to pick between the mean and a suitable alternative median for the simulated times. The median and mean are both advised in The Weibull distribution imputation, and the censors are the same for each of them. The median of the censored times becomes the same in the BS distribution since the distribution's median is equal to the scale parameter, as can be seen in Fig 10. The median is thus favored above the mean in the BS scenarios.

In the real data of the BC, the performance of The Weibull and BS distributions in the imputation of censored data was investigated. Based on the size of the DIC index for the BS distribution, it is smaller than The Weibull. In general, the BS distribution is more suitable for these data than the Weibull distribution.

In an article, Moghaddam et al. investigated the performance of the BA in the simulation of censoring times and analyzed its results with the Kaplan-Meier curve. In this study, the validity of the BA has been investigated by drawing different curves, and no numerical index has been used for decision-making. In this study, the distribution of failure times is considered the Weibull, although it may differ with the change in the distribution of the study results. One of the advantages of this study is the use of different censoring percentages and different sample sizes in examining the results. One of its weaknesses is not having an index to judge and using a distribution [7]. In our study, the mentioned weaknesses are covered.

Taylor et al. considered two approaches for multiple imputations of censored data. The distribution of participant at-risk times serves as the foundation for the data imputation approach, which is followed by point censoring. Therefore, the first step for both methods is to identify the distribution of times after censoring for those who are at risk for the event under study. The next step is the random selection of values for the imputation of censors. For this purpose, multiple imputations can be used instead of a selection. In the first method, called risk set imputation, for each censored time, a time is randomly selected from among the failure times after that point and is imputed instead of the censored time. This procedure continues up until the final instance of censoring. In the second method, called Kaplan-Meier Imputation, for each censoring time $t_j$, taking into account the times associated with people at risk $(Rj^+)$, a Kaplan-Meier survival curve $\hat{S}_{j+}(t)$ is considered. The imputation time value for each censoring $t^*(j)$ from cumulative distribution function $(1-\hat{S}_{j+}(t)$ and simulation from a uniform distribution $(U(0,1))$ and considering $t^*(j)$ associated with cumulative distribution function is selected [36]. In our study, 10,000 simulations have been performed for each censoring time, and it is possible to use different distributions, which is different from the mentioned study from this point of view.

Zhang et al. studied the imputation of data with interval censoring. In this study, which is used for HIV data, it is assumed that the date of infection diagnosis is interval-censored. For this purpose, two methods have been proposed. In the simple method, the middle interval is used for interval censoring. In the probability-based method, the interval censoring is considered as a discrete time interval with r value, and as $\boldsymbol{y}_i = (y_{i1}, \ldots, y_{ir_i}), i = 1, \ldots, n$, for each of the possible times in that interval, based on the selected distribution function, the probability is considered as $\boldsymbol{p}_i = (p_{i1}, \ldots, p_{ir_i})$. Assuming that $\boldsymbol{h}_i = (h_{i1}, \ldots, h_{ir_i})$ is the conditional probability of the $i_{\text{th}}$ person to have time $y_{ik}$, for each interval censoring time, their mean can be as $\boldsymbol{y}_i \boldsymbol{h}'_i$ or median as $y_i / h_i$ [37]. This study is valuable because it takes into account interval censoring as a method for imputation. However, given that one of the requirements of this method is to describe the bounds of interval censoring, we face a challenge.

In a study, Binbing et al. looked at the simulation of doubly censored data in survival data. In this study, the BA was used for the imputation of the censored. Based on this, in the first

step, the distribution of times is determined, and then a suitable prior is selected for the parameters of that distribution. In this paper, a multivariate normal distribution is suggested for the shape and scale parameters in a two-parameter distribution. The average of the simulated times was utilized for imputation [38], just like in our study. It should be mentioned that in our study, the use of the mean is also recommended in addition to the median, and the effectiveness of both indicators is assessed using a curve and an index. Testing the effectiveness of Bayesian simulation and the use of the median in competing distributions with the Weibull distribution is advised.

## 7- Conclusion

This study aims to look into the performance of the Bayesian approach in the context of censoring times. Considering that the result of the imputation is affected by the distribution of survival times, two practical distributions, The Weibull and Birnbaum-Sanders, were used. When the curves were used for comparison, the performance of the BA in the imputation of censoring times was evaluated to be similar to the two mentioned distributions. However using the DIC criteria, the BS distribution with a smaller index size works better than the Weibull distribution. It is also recommended to use the median of simulated times instead of their mean. In the future study, incomplete data will be analyzed with a BA and will be used to identify relationships between variables using a Bayesian network on a real dataset.

## Supporting information

**S1 Table. Values of the parameter $\theta$ of exponential distribution for different percentages of censoring and different values of parameters of Weibull distribution.**
(DOCX)

**S2 Table. Values of regression coefficients for different values of Weibull distribution shape parameter.**
(DOCX)

**S3 Table. Values of the parameter $\theta$ of exponential distribution for different percentages of censoring and different values of parameters of Birnbaum-Saunders distribution.**
(DOCX)

**S4 Table. Values of regression coefficients for different values of Birnbaum-Saunders distribution shape parameter.**
(DOCX)

**S1 Fig. Kaplan-Meier curves, Bayesian Approach(BA)(median of simulated times) curve under Weibull distribution, and failure times (Omitting-Censored) curve for each of the scenarios listed in S1 Table.** A) t~weibull(0.5,4), c~Exp(0.02), 10% censoring, and 100 sample sizes. B) t~weibull(0.5,4), c~exp(0.04), 20% censoring, and 100 sample sizes. C) t~weibull (0.5,4), c~Exp(0.30), 50% censoring, and 100 sample sizes. D) t~weibull(1,4), c~Exp(0.04), 10% censoring, and 100 sample sizes. E) t~weibull(1,4), c~exp(0.08), 20% censoring, and 100 sample sizes. F) t~weibull(1,4), c~Exp(0.30), 50% censoring, and 100 sample sizes. G) t~weibull(2,4), c~Exp(0.02), 10% censoring, and 100 sample sizes. H) t~weibull(2,4), c~exp(0.06), 20% censoring, and 100 sample sizes. I) t~weibull(2,4), c~Exp(0.20), 50% censoring, and 100 sample sizes.
(TIF)

**S2 Fig. Kaplan-Meier curves, Bayesian Approach(BA)(median of simulated times) curve under weibull distribution, and failure times (Omitting-Censored) curve for each of the**

**scenarios listed in S1 Table.** A) t~weibull(0.5,4), c~Exp(0.02), 10% censoring, and 300 sample sizes. B) t~weibull(0.5,4), c~exp(0.04), 20% censoring, and 300 sample sizes. C) t~weibull (0.5,4), c~Exp(0.30), 50% censoring, and 300 sample sizes. D) t~weibull(1,4), c~Exp(0.04), 10% censoring, and 300 sample sizes. E) t~weibull(1,4), c~exp(0.08), 20% censoring, and 300 sample sizes. F) t~weibull(1,4), c~Exp(0.30), 50% censoring, and 300 sample sizes. G) t~weibull(2,4), c~Exp(0.02), 10% censoring, and 300 sample sizes. H) t~weibull(2,4), c~exp(0.06), 20% censoring, and 300 sample sizes. I) t~weibull(2,4), c~Exp(0.20), 50% censoring, and 300 sample sizes.
(TIF)

**S3 Fig. Kaplan-Meier curves, Bayesian Approach(BA)(median of simulated times) curve under the Birnbaum-Saunders(BS) distribution, and failure times (Omitting-Censored) curve for each of the scenarios listed in S3 Table.** A) t~BS(0.5,4), c~Exp(0.01), 10% censoring, and 100 sample sizes. B) t~BS(0.5,4), c~exp(0.04), 20% censoring, and 100 sample sizes. C) t~BS(0.5,4), c~Exp(0.15), 50% censoring, and 100 sample sizes. D) t~BS(1,4), c~Exp(0.02), 10% censoring, and 100 sample sizes. E) t~BS(1,4), c~exp(0.04), 20% censoring, and 100 sample sizes. F) t~BS(1,4), c~Exp(0.15), 50% censoring, and 100 sample sizes. G) t~BS(2,4), c~Exp (0.01), 10% censoring, and 100 sample sizes. H) t~BS(2,4), c~exp(0.02), 20% censoring, and 100 sample sizes. I) t~BS(2,4), c~Exp(0.10), 50% censoring, and 100 sample sizes.
(TIF)

**S4 Fig. Kaplan-Meier curves, Bayesian Approach(BA)(median of simulated times) curve under Birnbaum-Saunders(BS) distribution, and failure times (Omitting-Censored) curve for each of the scenarios listed in S3 Table.** A) t~BS(0.5,4), c~Exp(0.03), 10% censoring, and 300 sample sizes. B) t~BS(0.5,4), c~Exp(0.05), 20% censoring, and 300 sample sizes. C) t~BS (0.5,4), c~Exp(0.15), 50% censoring, and 300 sample sizes. D) t~BS(1,4), c~Exp(0.02), 10% censoring, and 300 sample sizes. E) t~BS(1,4), c~Exp(0.04), 20% censoring, and 300 sample sizes. F) t~BS(1,4), c~Exp(0.15), 50% censoring, and 300 sample sizes. G) t~BS(2,4), c~Exp (0.01), 10% censoring, and 300 sample sizes. H) t~BS(2,4), c~Exp(0.02), 20% censoring, and 300 sample sizes. I) t~BS(2,4), c~Exp(0.15), 50% censoring, and 300 sample sizes.
(TIF)

**S1 File. WibBUGS and R codes.**
(DOCX)

# Acknowledgments

Hereby, I thank to colleague of the biostatistics department of Tarbiat Modares University and the Cancer Research Center of Shahid Beheshti University of Medical Sciences for their guidance and providing valuable comments. This paper is part of a thesis in the Biostatistics Department of Tarbiat Modares University, Tehran, Iran.

# Author Contributions

**Conceptualization:** Parviz Shahmirzalou, Aliakbar Rasekhi, Majid Jafari Khaledi, Maryam Khayamzadeh.

**Data curation:** Parviz Shahmirzalou, Aliakbar Rasekhi, Maryam Khayamzadeh.

**Formal analysis:** Parviz Shahmirzalou, Aliakbar Rasekhi, Majid Jafari Khaledi, Maryam Khayamzadeh.

**Investigation:** Parviz Shahmirzalou, Majid Jafari Khaledi, Maryam Khayamzadeh.

**Methodology:** Parviz Shahmirzalou, Aliakbar Rasekhi, Majid Jafari Khaledi.

**Project administration:** Parviz Shahmirzalou.

**Resources:** Parviz Shahmirzalou.

**Software:** Parviz Shahmirzalou, Majid Jafari Khaledi.

**Supervision:** Parviz Shahmirzalou, Aliakbar Rasekhi, Maryam Khayamzadeh.

**Validation:** Parviz Shahmirzalou, Aliakbar Rasekhi, Majid Jafari Khaledi, Maryam Khayamzadeh.

**Visualization:** Parviz Shahmirzalou, Aliakbar Rasekhi, Majid Jafari Khaledi, Maryam Khayamzadeh.

**Writing – original draft:** Parviz Shahmirzalou.

**Writing – review & editing:** Parviz Shahmirzalou.

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
