## [Decision Letter · Decision Letter 0]

24 Oct 2023

PONE-D-23-28410Comparison of the Performance of Bayesian Approach with Weibull and Birnbaum-Saunders Distributions in Imputation of Time-to-Event CensorsPLOS ONE

Dear Dr. Rasekhi,

Thank you for submitting your manuscript to PLOS ONE. After careful consideration, we feel that your manuscript will likely be suitable for publication if it is revised to address the points below. Therefore, my decision is "Major Revision".

We look forward to receiving your revised manuscript.

Kind regards,

Oluwafemi Samson Balogun, Ph.D.

Academic Editor

PLOS ONE

Journal Requirements:

6. We note you have included a table to which you do not refer in the text of your manuscript. Please ensure that you refer to Table 4 in your text; if accepted, production will need this reference to link the reader to the Table.

Reviewers' comments:

Reviewer's Responses to Questions

**Comments to the Author**

1. Is the manuscript technically sound, and do the data support the conclusions?

Reviewer #1: Partly

Reviewer #2: Partly

2. Has the statistical analysis been performed appropriately and rigorously? 

Reviewer #1: Yes

Reviewer #2: Yes

3. Have the authors made all data underlying the findings in their manuscript fully available?

Reviewer #1: No

Reviewer #2: No

4. Is the manuscript presented in an intelligible fashion and written in standard English?

Reviewer #1: No

Reviewer #2: No

5. Review Comments to the Author

Reviewer #1: This study aimed to investigate the performance of the Bayesian Approach (BA) in the imputation of censored records in simulated and Breast Cancer (BC) data. Due to the difference in the distribution of time to the occurrence of the event in survival analysis, two well-known Weibull and Birnbaum-Saunders (BS) distributions have been used to test the performance of the BA. For each of the censored, 10.000 times were simulated using the BA in R and BUGS software, and their median or mean was imputed instead of each censor. The eligibility of both imputation methods was investigated using different curves, different censoring percentages, and sample sizes, as well as the Deviance Information Criteria (DIC) index in simulated data and especially real BC data. The BC data contains 220 patients who were identified and followed up between 2015 to 2023, was made accessible on February 1st, 2023. The Kaplan-Meier curve, the BA and survival curve were drawn for the observed times. Findings showed that the

performance of the BA under the Weibull and BS distributions in simulated data is similar. The DIC index in BC data under the BS distribution (1510) is less than the Weibull distribution (1698). Therefore, the BS distribution is preferred over the Weibull for imputation of censoring times.

On page 4, line 113, what is meant by 4ecommend Pratama et al. ?

References are not correctly cited. Please cite them carefully.

Equations are not written properly. For example, the median survival equation does not looks correct.

To determine scale parameter, how the values of x are generated?

Weibull must be written with capital W.

I still cannot figure out why imputation of censored data because censoring mechanism is different from missing value? Why we have survival analysis to deal censored data if we impute them?

Figures are too messy and cannot be readable. Improve the presentation of figures.

I tried to run the code but it did not work. Please recheck the code.

Include effective size as the diagnostic tool.

Still I cannot figured out how the Bayesian analysis is done?

Reviewer #2: This study aimed to investigate the performance of the Bayesian Approach (BA) in the imputation of censored records in simulated and Breast Cancer (BC) data. Due to the difference in the distribution of time to the occurrence of the event in survival analysis, two well-known Weibull and Birnbaum-Saunders (BS) distributions have been used to test the performance of the BA.

References are not correctly cited.

Equations are also not written properly. For example, the median survival equation.

The authors must give comprehensive details on how the scale parameter is determined and how the values of X are generated.

Weibull must be written with capital W.

Why do we have survival analysis to deal censored data if we impute them?

Figures are not readable.

Why gamma distribution is used as a prior?

How the DIC values are computed from the MCMC output?

Some graphical assessment tools for MCMC convergence must be included.

6. PLOS authors have the option to publish the peer review history of their article (what does this mean?). If published, this will include your full peer review and any attached files.

Reviewer #1: No

Reviewer #2: No

---

## [Author Response · Author response to Decision Letter 0]

30 Nov 2023

Reviewer #1:

1. Have the authors made all data underlying the findings in their manuscript fully available?

Author’s response: 

Thanks.

The code and Breast cancer data have been uploaded.

GitHub Link for R codes:

https://github.com/pshkhoei/Thesis-Paper-1

2. Is the manuscript presented in an intelligible fashion and written in standard English?

Author’s response: 

Many thanks.

The article has been thoroughly reviewed and modified. Grammar and spelling mistakes were fixed.

3. On page 4, line 113, what is meant by 4ecommend Pratama et al. ?

Author’s response: 

Thank you very much. Corrected.

4. References are not correctly cited. Please cite them carefully.

Author’s response: 

Thanks. All of the references were double-checked, and any necessary changes were made.

5. Equations are not written properly. For example, the median survival equation does not look correct.

Author’s response:

Many thanks. 

Every formula, particularly the median survival time, has been reviewed and updated.

6. To determine scale parameter, how the values of x are generated? 

Author’s response: 

Thanks a lot. 

There are two possible states for the variable x: zero and one, with equal probability of 0.5. This variable is used in the scale parameter of the distributions of the survival time (Weibull and Birnbaum-Saunders). That is, instead of using a fixed values (say 4) for the scale parameter, the scale was linked to a two-state variable which represented two groups.

7. Weibull must be written with capital W. 

Author’s response: 

Thank you very much. Corrected.

8. I still cannot figure out why imputation of censored data because censoring mechanism is different from missing value? 

Author’s response: 

Thank you. 

The Bayesian approach can be used to impute missing and censored data. The aim of this research is to apply the Bayesian method to impute censoring times. For this purpose, the computations in Sections 4.1 to 4.3 are written for censored times conditionally. Review, for instance, equations 5 through 7.

 f(t│T≥c,θ)= (f(t│θ))/(S(c│θ) )

E(T│T≥c)= (∫_c^∞▒〖t f(t│θ)dt〗)/(S(c│θ) )

 0.5= S(T|T≥c,θ) =(P(T≥t_med |θ))/(P(T≥c|θ))=(S(t_med |θ))/(S(c|θ))

If imputation is used to fill in missing data, the density function, mean, and median survival time should be written in the aforementioned equations unconditionally. 

9. Why we have survival analysis to deal censored data if we impute them? 

Author’s response:

Regards. Imputation can be used for a number of purposes.

 Survival models are utilized when the dependent variable is censored itself.

 If censoring is documented in other variables, it is required to impute the censored ones (See Reference 1 below). For example, in cancer patients, the dependent variable is survival time. However, disease recurrence time is an independent variable with interval censoring. As a result, censorship must be considered, and censored data should be imputed while analyzing this variable.

 When we aim to employ a method other than survival analysis. For example, in the Structural Equation Modeling (SEM) and the Bayesian network, both of which need the imputation of censored data. In such circumstances, the censored data must be managed in some way in order to have complete the data.

 When we wish to perform exploratory analysis.

 Assume we want to use an independent t-test to compare the average survival time of men and women. Assuming that part of the survival times is censored, comparisons on incomplete survival times may yield incorrect results. The same is true for Analysis of Variance (ANOVA) (See Reference 2 below).

10. Figures are too messy and cannot be readable. Improve the presentation of figures.

Author’s response: 

Many thanks

All plots were redesigned, and high-quality diagrams were created. To avoid losing the quality of the graphs, the output of the R program was saved as a pdf.

11. I tried to run the code but it did not work. Please recheck the code. 

Author’s response: 

Thanks.

The code and Breast cancer data have been uploaded.

GitHub Link for R codes:

https://github.com/pshkhoei/Thesis-Paper-1

12. Include effective size as the diagnostic tool. 

Author’s response: 

Thanks a lot. Done.

13. Still I cannot figured out how the Bayesian analysis is done?

Author’s response: 

Thankful.

The following steps are taken.

1- A distribution is assigned to the desired censored variable (Weibull or Birnbaum-Sanders).

2-The prior distribution is assigned to the parameters of the distribution.

3- Under the Bayesian approach, the value of the parameters is updated 10,000 times using the likelihood function and prior distribution (the posterior distribution of the parameters (p(θ│D)) is estimated).

4- With Equation (7) in the paper:

f(t│T≥c,D)= ∫▒〖f(t│T≥c,θ)p(θ│D)dθ〗

each value of the posterior parameter is entered into Equation (7), and one data point is generated for each censored data point. If the posterior parameters are estimated 10,000 times, the censored value is simulated 10,000 times.

5- The censored data is replaced with the mean or median of the simulated times to have complete data.

6- The desired graph or analysis is performed based on the entire set of data.

References

1. Svahn C, Sysoev O. Selective Imputation of Covariates in High Dimensional Censored Data. Journal of Computational and Graphical Statistics. 2022;31(4):1397-405.

2. Fan J. Imputation Based Statistical Test for Right Censored Data. Kentucky: University of Louisville; 2007.

Reviewer #2

1. References are not correctly cited. 

Author’s response: 

Thanks.

All of the references were double-checked, and any necessary changes were made.

2.Equations are also not written properly. For example, the median survival equation.

Author’s response: 

Many thanks.

The article has been thoroughly reviewed and modified. Grammar and spelling mistakes were fixed.

3. The authors must give comprehensive details on how the scale parameter is determined and how the values of X are generated. 

Author’s response: 

Thanks a lot. 

There are two possible states for the variable x: zero and one, with equal probability of 0.5. This variable is used in the scale parameter of the distributions of the survival time (Weibull and Birnbaum-Saunders). That is, instead of using a fixed values (say 4) for the scale parameter, the scale was linked to a two-state variable which represented two groups.

4. Weibull must be written with capital W. 

Author's response: 

Thanks a lot. Done

5. Why do we have survival analysis to deal censored data if we impute them? 

Author’s response:

Regards. Imputation can be used for a number of purposes.

 Survival models are utilized when the dependent variable is censored itself.

 If censoring is documented in other variables, it is required to impute the censored ones (See Reference 1 below). For example, in cancer patients, the dependent variable is survival time. However, disease recurrence time is an independent variable with interval censoring. As a result, censorship must be considered, and censored data should be imputed while analyzing this variable.

 When we aim to employ a method other than survival analysis. For example, in the Structural Equation Modeling (SEM) and the Bayesian network, both of which need the imputation of censored data. In such circumstances, the censored data must be managed in some way in order to have complete the data.

 When we wish to perform exploratory analysis.

 Assume we want to use an independent t-test to compare the average survival time of men and women. Assuming that part of the survival times is censored, comparisons on incomplete survival times may yield incorrect results. The same is true for Analysis of Variance (ANOVA) (See Reference 2 below).

6. Figures are not readable. 

Author’s response: 

Many thanks.

All plots were redesigned, and high-quality diagrams were created. To avoid losing the quality of the graphs, the output of the R program was saved as a pdf.

7. Why gamma distribution is used as a prior? 

Author's response:

Many thanks. Other distributions can also be used for prior distribution of parameters. In this study, the gamma distribution is used and its parameters are chosen such that we have a reference prior to provide a common base to evaluate data, rather than specific prior information.

8. How the DIC values are computed from the MCMC output? 

Author's response: 

Regards. An example is provided to clarify this question. 

We define y as having a normal distribution with a mean (θ) and a precision parameter (τ).

y_1,…,y_n |θ~N(θ,1/τ^* ),

 For the desired parameter, the likelihood function and prior distribution are represented as L(θ|y)∝exp⁡(-(nτ^*)/2 (y ®-θ)^2), and θ~N(θ_0,1/τ_0 ), respectively, and the posterior distribution of the parameter is also expressed as follows.

N(τ_0/(τ_0+nτ_* ) θ_0+ (nτ_*)/(τ_0+nτ_* ) (y.) ®, 1/(τ_0+nτ_* ))

Our task is to compute the DIC index.

DIC=2 E(D(θ│y))-D((θ)) ^ (1)

As a result, E(D(θ│y)) and D((θ)) ^ must be defined.

 D(θ)=nτ_* ((y.) ®-θ )^2, (2)

 D(θ ^ )=nτ_* ((y.) ®-θ ^ )^2 (3)

 θ ^=E(θ|y) (4)

 D(θ ^ )=nτ_* ((y.) ®-E(θ|y )^2 (5)

 We must now compute E(D(θ│y).

E(D(θ│y)=E(nτ_* ((y.) ®-θ )^2 |y)= nτ_* E(((y.) ®-θ )^2│y)= nτ_* Var(θ│y)+nτ_* (E(θ│y))^2=

nτ_* Var(θ│y)+nτ_* (θ ^- y ® )^2= (nτ_*)/(nτ_*+ τ_0 )+D(θ ^ )= p_D+ D(θ ^ ) (6)

And we rewrite D(θ ^ ) using equation 5.

D(θ ^ )=nτ_* ((y.) ®-E(θ|y )^2= nτ_* ((y.) ®-[(nτ_*)/(τ_0+ nτ_* ) (y.) ®+ τ_0/(τ_0+ nτ_* ) θ_0 ])^2= 

nτ_* (τ_0/(τ_0+ nτ_* ))^2 〖(y ®.- θ_0)〗^2, (7)

The DIC index is expressed as follows using relations (1), (6), and (7).

DIC= 2 E(D(θ│y))-D((θ)) ^=2 p_D+ 2 D(θ ^ )- D(θ ^ )= 2 p_D+ D(θ ^ )=

= 2 (nτ_*)/(nτ_*+ τ_0 )+ nτ_* (τ_0/(τ_0+ nτ_* ))^2 〖(y ®.- θ_0)〗^2, (8)

Consequently, equation (8), and (3) is used to get the value of DIC (See Reference 3 below). The size of the DIC index is obtained by using the corresponding command code.

9. Some graphical assessment tools for MCMC convergence must be included.

Author's response:

Thanks a lot. I appreciate the point you raised. In this regard, the convergence of the simulations was evaluated using the Geweke’s diagnostics and is shown in graphs (6) and (7). The Geweke’s diagnostic Z-score indicate that the convergence is satisfied.

References

1. Svahn C, Sysoev O. Selective Imputation of Covariates in High Dimensional Censored Data. Journal of Computational and Graphical Statistics. 2022;31(4):1397-405.

2. Fan J. Imputation Based Statistical Test for Right Censored Data. Kentucky: University of Louisville; 2007.

3. Christensen, R., Johnson, W., Branscum, A., & Hanson, T. E. (2010). Bayesian ideas and data analysis: an introduction for scientists and statisticians. CRC press.

---

## [Decision Letter · Decision Letter 1]

4 Dec 2023

Comparison Performance of the Bayesian Approach with the Weibull and Birnbaum-Saunders Distributions in Imputation of Time-to-Event Censors

PONE-D-23-28410R1

Dear Dr. Aliakbar Rasekhi,

We’re pleased to inform you that your manuscript has been judged scientifically suitable for publication and will be formally accepted for publication once it meets all outstanding technical requirements.

Kind regards,

Oluwafemi Samson Balogun, Ph.D.

Academic Editor

PLOS ONE

Additional Editor Comments (optional):

Reviewers' comments:

Reviewer's Responses to Questions

**Comments to the Author**

1. If the authors have adequately addressed your comments raised in a previous round of review and you feel that this manuscript is now acceptable for publication, you may indicate that here to bypass the “Comments to the Author” section, enter your conflict of interest statement in the “Confidential to Editor” section, and submit your "Accept" recommendation.

Reviewer #1: All comments have been addressed

Reviewer #2: All comments have been addressed

2. Is the manuscript technically sound, and do the data support the conclusions?

Reviewer #1: Partly

Reviewer #2: Yes

3. Has the statistical analysis been performed appropriately and rigorously? 

Reviewer #1: Yes

Reviewer #2: Yes

4. Have the authors made all data underlying the findings in their manuscript fully available?

Reviewer #1: Yes

Reviewer #2: Yes

5. Is the manuscript presented in an intelligible fashion and written in standard English?

Reviewer #1: Yes

Reviewer #2: Yes

6. Review Comments to the Author

Reviewer #1: This study aimed to investigate the performance of the Bayesian Approach (BA) in the imputation of censored records in simulated and Breast Cancer (BC) data. Due to the difference in the distribution of time to the occurrence of the event in survival analysis, two well-known Weibull and Birnbaum-Saunders (BS) distributions have been used to test the performance of the BA. For each of the censored, 10.000 times were simulated using the BA in R and BUGS software, and their median or mean was imputed instead of each censor. The eligibility of both imputation methods was investigated using different curves, different censoring percentages, and sample sizes, as well as the Deviance Information Criteria (DIC) index in simulated data and especially real BC data. The BC data contains 220 patients who were identified and followed up between 2015 to 2023, was made accessible on February 1st, 2023. The Kaplan-Meier curve, the BA and survival curve were drawn for the observed times. Findings showed that the

performance of the BA under the Weibull and BS distributions in simulated data is similar. The DIC index in BC data under the BS distribution (1510) is less than the Weibull distribution (1698). Therefore, the BS distribution is preferred over the Weibull for imputation of censoring times.

The previous queries are addressed.

Reviewer #2: The authors addressed all my concerns. Hence, I recommend the paper for publication in its present form.

7. PLOS authors have the option to publish the peer review history of their article (what does this mean?). If published, this will include your full peer review and any attached files.

Reviewer #1: No

Reviewer #2: No

---

## [Editor Report · Acceptance letter]

11 Jan 2024

PONE-D-23-28410R1 

PLOS ONE

Dear Dr. Rasekhi, 

I'm pleased to inform you that your manuscript has been deemed suitable for publication in PLOS ONE. Congratulations! Your manuscript is now being handed over to our production team.

Kind regards, 

on behalf of

Dr. Oluwafemi Samson Balogun 

Academic Editor

PLOS ONE